# Predictive value of abnormal blood tests for detecting cancer in primary care patients with nonspecific abdominal symptoms: A population-based cohort study of 477,870 patients in England

**Meena Rafiq** [1,2]*, **Cristina Renzi** [1,3], **Becky White** [1], **Nadine Zakkak** [1], **Brian Nicholson** [4], **Georgios Lyratzopoulos** [1], **Matthew Barclay** [1]

1 Epidemiology of Cancer Healthcare & Outcomes (ECHO) Group, Department of Behavioural Science, Institute of Epidemiology and Health Care (IEHC), UCL, London, United Kingdom, 2 Department of General Practice and Primary Care, Centre for Cancer Research, University of Melbourne, Melbourne, Australia, 3 Faculty of Medicine, University Vita-Salute San Raffaele, Milan, Italy, 4 Nuffield Department of Primary Care Health Sciences, University of Oxford, Oxford, United Kingdom

* Meena.rafiq@ucl.ac.uk

## Abstract

### Background

Identifying patients presenting with nonspecific abdominal symptoms who have underlying cancer is a challenge. Common blood tests are widely used to investigate these symptoms in primary care, but their predictive value for detecting cancer in this context is unknown. We quantify the predictive value of 19 abnormal blood test results for detecting underlying cancer in patients presenting with 2 nonspecific abdominal symptoms.

### Methods and findings

Using data from the UK Clinical Practice Research Datalink (CPRD) linked to the National Cancer Registry, Hospital Episode Statistics and Index of Multiple Deprivation, we conducted a population-based cohort study of patients aged ≥30 presenting to English general practice with abdominal pain or bloating between January 2007 and October 2016. Positive and negative predictive values (PPV and NPV), sensitivity, and specificity for cancer diagnosis (overall and by cancer site) were calculated for 19 abnormal blood test results co-occurring in primary care within 3 months of abdominal pain or bloating presentations.

A total of 9,427/425,549 (2.2%) patients with abdominal pain and 1,148/52,321 (2.2%) with abdominal bloating were diagnosed with cancer within 12 months post-presentation. For both symptoms, in both males and females aged ≥60, the PPV for cancer exceeded the 3% risk threshold used by the UK National Institute for Health and Care Excellence for recommending urgent specialist cancer referral. Concurrent blood tests were performed in two thirds of all patients (64% with abdominal pain and 70% with bloating). In patients aged 30 to 59, several blood abnormalities updated a patient's cancer risk to above the 3% threshold:

**Data Availability Statement:** This study is based in part on data from the Clinical Practice Research Datalink obtained under licence from the UK Medicines and Healthcare products Regulatory Agency. The data is provided by patients and

collected by the NHS as part of their care and support. The interpretation and conclusions contained in this study are those of the author/s alone. Code is available from https://osf.io/52g9d/. Due to privacy laws and the data user agreement between UCL and CPRD, researchers are not authorised to share individual patient data. Access to CPRD data, including UK Primary Care Data, and linked data such as Hospital Episode Statistics, is subject to protocol approval via CPRD's Research Data Governance (RDG) Process, see https://cprd. com/data-access for further details.

**Funding:** The study was supported by RREDD-EHR project supported by the International Alliance for Cancer Early Detection (ACED) a partnership between Cancer Research UK (CRUK) (C18081/A31373), Canary Center at Stanford University, the University of Cambridge, OHSU Knight Cancer Institute, University College London, and the University of Manchester. GL acknowledges an Advanced Clinician Scientist Fellowship from CRUK (C18081/A18180). CR acknowledges funding from CRUK (grant number EDDCPJT\100018). BN acknowledges funding from a National Institute of Health Research (NIHR) Academic Clinical Lectureship, a CRUK Research Careers Committee Postdoctoral Fellowship (RCCPDF/100005) and is the Early Detection Theme Lead for the CRUK Oxford Cancer Centre (CTRQQR-2021\100002). MB acknowledges an ACED Pathway Award (CRUK EDDAPA-2022/100002). The funders had no role in study design, data collection and analysis, decision to publish, or preparation of the manuscript.

**Competing interests:** We declare no support from any organisation for the submitted work. BDN receives institutional research funding from GRAIL Inc unrelated to this study, MB receives personal fees from GRAIL Inc for Independent Data Monitoring Committee (IDMC) membership unrelated to this study and BW received personal funding from the British Liver Trust in 2023 for an unrelated data consultancy project, otherwise there are no financial relationships with any organisations that might have an interest in the submitted work in the previous three years and no other relationships or activities that could appear to have influenced the submitted work.

**Abbreviations:** ALP, alkaline phosphatase; ALT, alanine transaminase; CI, confidence interval; CPRD, Clinical Practice Research Datalink; CRP, C reactive protein; ESR, erythrocyte sedimentation rate; FDS, Faster Diagnosis Standard; GP, general practitioner; HES APC, Hospital Episode Statistics Admitted Patient Care; IMD, index of multiple deprivation; ISAC, Independent Scientific Advisory Committee; NPV, negative predictive value;

For example, in females aged 50 to 59 with abdominal bloating, pre-blood test cancer risk of 1.6% increased to: 10% with raised ferritin, 9% with low albumin, 8% with raised platelets, 6% with raised inflammatory markers, and 4% with anaemia. Compared to risk assessment solely based on presenting symptom, age and sex, for every 1,000 patients with abdominal bloating, assessment incorporating information from blood test results would result in 63 additional urgent suspected cancer referrals and would identify 3 extra cancer patients through this route (a 16% relative increase in cancer diagnosis yield). Study limitations include reliance on completeness of coding of symptoms in primary care records and possible variation in PPVs if extrapolated to healthcare settings with higher or lower rates of blood test use.

## Conclusions

In patients consulting with nonspecific abdominal symptoms, the assessment of cancer risk based on symptoms, age and sex alone can be substantially enhanced by considering additional information from common blood test results. Male and female patients aged ≥60 presenting to primary care with abdominal pain or bloating warrant consideration for urgent cancer referral or investigation. Further cancer assessment should also be considered in patients aged 30 to 59 with concurrent blood test abnormalities. This approach can detect additional patients with underlying cancer through expedited referral routes and can guide decisions on specialist referrals and investigation strategies for different cancer sites.

## Author summary

### Why was this study done?

- Half of all patients with as-yet-undetected cancer will first present with nonspecific symptoms that can be challenging to diagnose.

- Many of these patients are investigated in primary care with commonly used blood tests that could help to identify which patients are most likely to have underlying cancer (to prioritise them for referral) and which patients can be safely monitored in primary care.

- This study aimed to assess the predictive value of abnormal blood tests for detecting cancer in patients presenting to primary care with 2 nonspecific abdominal symptoms.

### What did the researchers do and find?

- Using linked UK primary care data (CPRD), we conducted a cohort study of 477,870 patients aged ≥30 years presenting with new abdominal pain or bloating and calculated the predictive value of 19 abnormal blood test results for detecting cancer by age and sex.

- Males and females aged ≥60 presenting with either symptom had a risk of underlying cancer exceeding the 3% threshold used by the UK National Institute for Health and Care Excellence for recommending urgent cancer referral.

NRESC, National Research Ethics Service Committee; ONS, office for national statistic; PPI, patient and public involvement; PPV, positive predictive value; PSA, prostate specific antigen; WBC, white blood cell.

- In patients aged 30 to 59 with abdominal pain or bloating, several blood abnormalities updated a patient's cancer risk to above the 3% threshold and they should be considered for urgent cancer referral.

### What do these findings mean?

- Commonly used primary care blood test results can improve the detection of underlying cancer in patients consulting with nonspecific abdominal symptoms.

- These findings can inform updates to clinical guidelines to allow detection of additional patients with underlying cancer through expedited referral routes and can guide decisions on specialist referrals and investigation strategies for different cancer sites.

- Limitations include the results applying to patients who had been recorded as having abdominal pain and bloating by their clinician and who had been selected by the clinician for blood testing (and therefore have a higher cancer risk that all patients with abdominal pain and bloating).

## Introduction

Abdominal symptoms account for 10% of all primary care consultations [1]. Every month 1 in 10 people over the age of 50 will experience abdominal pain and a quarter of these patients will consult their general practitioner (GP) [2]. Most nonspecific abdominal symptoms, like pain or bloating, have a benign cause [3,4], but in a small proportion of patients they are features of undiagnosed cancer of different sites [5–7]. Referring all patients with abdominal symptoms is not recommended and would result in many unnecessary investigations. Identifying which patients with nonspecific abdominal symptoms are at highest risk of cancer so they can be prioritised for further investigation is an ongoing challenge.

Faced with this challenge, policymakers such as the UK's National Institute for Health and Care Excellence (NICE) have issued guidelines that recommend urgent cancer referral in patients with a cancer risk of ≥3% (to be made at the time of first clinical presentation with features of suspected cancer) and set the Faster Diagnosis Standard (FDS) specifying that a definitive diagnosis or ruling out of cancer should occur within 28 days of referral [8]. For abdominal pain and bloating, this includes patients in older age groups only if other symptoms (weight loss, nausea and vomiting, rectal bleeding) or blood test abnormalities (anaemia or raised platelets) are present [9]. Similar guidelines exist in Denmark [10], Spain [11], Sweden [12], and Norway [13], among other European countries, and Australia [14]. However, typically within current guidelines they predominantly focus on the presence of "alarm" symptoms and risk of cancer of a single organ, with each cancer site having different recommended investigations. Limited guidance exists for the management of nonspecific symptoms or the relative cancer risk of different cancers to guide investigation strategy and referral decisions.

Despite many blood test abnormalities being associated with cancer [15], and most patients with nonspecific symptoms who have undiagnosed cancer being investigated with these tests in primary care [16], only information from anaemia and raised platelets are currently included in recommendations for patients with nonspecific abdominal symptoms. The limits

of current referral guidelines are in part due to little evidence existing on the value of abnormal blood tests for supporting cancer risk assessment in patients presenting with these symptoms. Such abnormalities can be particularly relevant in patients at cancer risk levels close to the recommended referral risk thresholds based on their symptomatic presentation alone. Blood test results could help reclassify patients with abnormal results to a group with an elevated posttest risk in need of further cancer investigation, while those with normal blood test results and lower posttest risk can be managed without a specialist referral. Additionally, abnormal blood test results could identify cancer sites that should be prioritised for further cancer assessment to optimise investigation strategies and specialist referral decisions.

We aimed to use linked primary care data to quantify the predictive value of 19 abnormal blood test results for detecting underlying cancer in patients presenting with 2 nonspecific abdominal symptoms. We focused on predictive values exceeding the 3% risk threshold set by NICE for recommending urgent cancer assessment. This knowledge could inform clinical practice guidelines both in the UK and internationally.

## Methods

### Ethical approval

The study was approved by the MHRA (UK) Independent Scientific Advisory Committee (ISAC) (protocol number: 18_299R), under Section 251 (NHS Social Care Act 2006). Generic ethical approval for observational studies conducted using anonymised CPRD data with approval from ISAC has been granted from a National Research Ethics Service Committee (NRESC). The study was performed in accordance with the Declaration of Helsinki.

### Study design and setting

A population-based cohort study using primary care data from the UK Clinical Practice Research Datalink (CPRD) [17] linked at the person-level to English National Cancer Registry [18], Hospital Episode Statistics Admitted Patient Care (HES APC) [19], and index of multiple deprivation (IMD) datasets. In the UK healthcare system GPs act as gatekeepers to secondary care and primary care is the setting where most symptomatic patients with underlying cancer first present [20]. CPRD contains anonymised electronic primary care records from approximately 9% of all UK practices [17], including coded information on patient consultations, laboratory results, and demographics (the analysis plan can be found in the S1 Supplementary File).

### Study population

This study involved 2 different study populations. Firstly, a population of patients who present in primary care with new abdominal pain and bloating (the symptomatic population); this population was used to estimate the predictive value of each symptom (considered in isolation) for detecting underlying cancer in the following year. Secondly, a subgroup of the above population, comprising symptomatic patients (as above) who also had blood tests around the time of their presentation with the studies symptoms (the "tested" symptomatic population); this population was used to the estimate the predictive value of symptoms + blood tests for cancer in the following year. These 2 study populations were identified as follows.

### Defining the symptomatic population

Patients registered at a CPRD practice, aged ≥30 years, and with a new primary care record of either abdominal pain or bloating between 1st January 2007 and 31st October 2016 were identified. This age group was selected as cancer risk is considerably lower in patients aged <30

years; moreover, cancer aetiology and cancer-site case mix different substantially, meriting a different, bespoke, study explicitly addressing cancer risk in symptomatic adolescents and younger adults. A new presentation was defined as a coded episode of abdominal pain or bloating (see Table A in S1 Supplementary File for code lists) during this period with no other coded episode of the same symptom in the preceding 365 days. For each patient, the dates of each eligible symptom episode during the study period were recorded (symptom date). The first eligible episode of each symptom was selected (index date) and used to create 2 cohorts (abdominal pain and abdominal bloating). Patients were excluded if they had not been registered at a CPRD practice with up to standard data quality for over a year, were not eligible for linkage to the national cancer registry dataset, or if they had a cancer diagnosis recorded in CPRD/HES APC/the national cancer registry dataset before their index date.

## Defining the "tested" symptomatic population

Sixteen commonly used blood tests were selected a priori for inclusion in this study based on clinical knowledge and the existing literature [15,21–41]. They comprised tumour markers (prostate specific antigen (PSA), CA125); acute phase reactants (platelets, erythrocyte sedimentation rate (ESR), C reactive protein (CRP), ferritin, and total white blood cell count (WBC)); markers of iron deficiency or anaemia (ferritin and haemoglobin); liver, renal, or bone profile tests (bilirubin, albumin, AST (aspartate aminotransferase), ALT (alanine transaminase), ALP (alkaline phosphatase), calcium, and creatinine); and glycosylated haemoglobin (HbA1c) (see Table B in S1 Supplementary File for code lists and evidence/rationale for inclusion of blood tests).

Only patients who had primary care blood tests occurring around the time of symptomatic presentation were included in the "tested" symptomatic population. To avoid introducing immortal time bias, 2 subgroups of symptomatic patients who had blood tests were created (Fig A in S1 Supplementary File) by identifying the following scenarios:

- Scenario 1: Blood test followed by symptom presentation within 3 months (in this scenario, the later symptom date = index date).

- Scenario 2: Symptom presentation followed by blood test within 3 months (in this scenario, the date of the last blood test request in this period = index date).

For each patient, the first identified symptom-blood test event (and corresponding index date) was selected to make the final study cohorts: "tested abdominal pain patients" and "tested abdominal bloating patients" (Fig 1). Data were extracted from CPRD on the date and result of all relevant blood tests in the 3 months before the identified index date. Results with biologically implausible values were excluded and duplicates were removed (selecting the mean of the result if values differed). Each result was classified as normal or abnormal based on standard laboratory reference ranges.

## Defining the outcome

Each patient was followed up for 12 months after their index date to identify any new cancer diagnoses (diagnosis date and site) coded using ICD-10 codes in the national cancer registry dataset (excluding non-melanoma skin cancer and benign brain tumours, see Table C in S1 Supplementary File for code list). Twelve months was selected based on analysis of how long cancer incidence remained elevated following new abdominal pain or bloating presentations (Fig B in S1 Supplementary File).

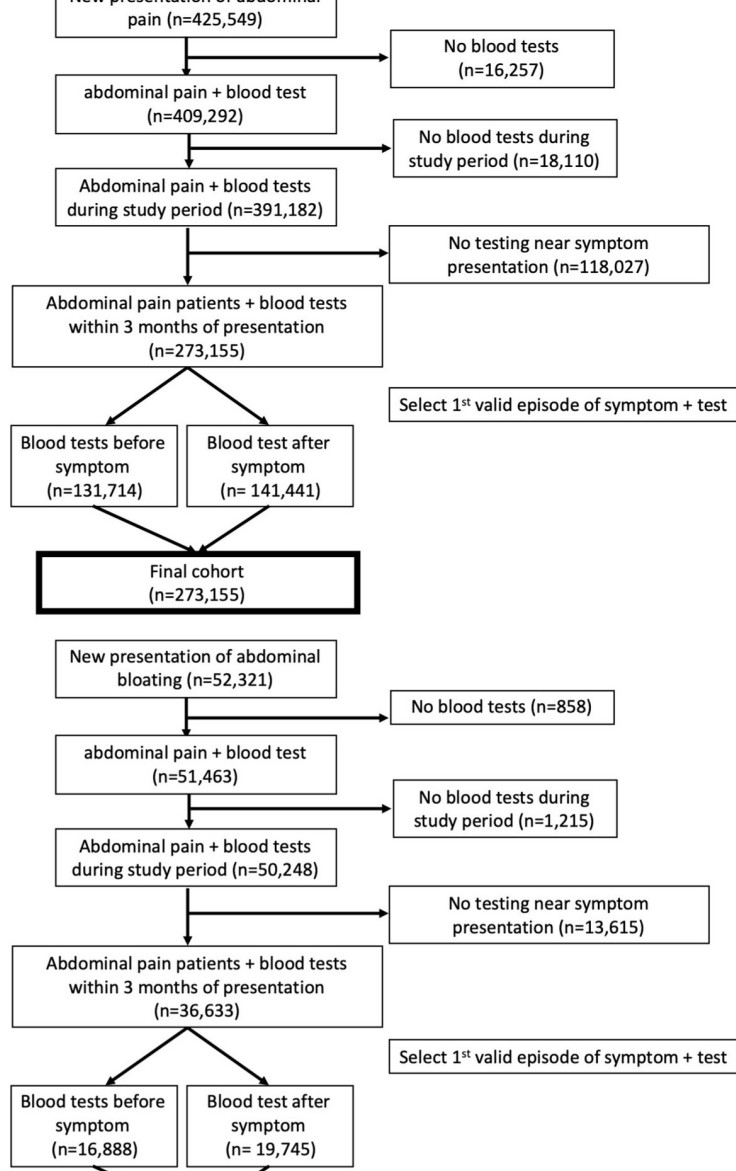

**Fig 1. Flowchart outlining the steps in creating the final cohort of tested abdominal pain (top panel) and tested abdominal bloating (bottom panel) patients.**

### Patient and public involvement (PPI)

The findings of this study were presented at 2 PPI meetings to share the findings and receive feedback informing the development of future research proposals expanding the research, specifically regarding the research question, objectives, and proposed outcomes. These meetings took the format of online focus groups with guided discussion points involving 11 participants with lived experience of cancer.

## Statistical analysis

### Symptomatic cohort

Chi-squared tests were used to compare the baseline characteristics of the 2 symptomatic cohorts (abdominal pain and bloating). The distribution of cancer diagnoses by site and sex was then compared for each symptom cohort. For each symptom, the 1-year cancer incidence was calculated from the first symptom date, stratifying by sex. This is equivalent to the positive predictive value (PPV) of abdominal pain or abdominal bloating for cancer, also referred to as the cancer risk [22]. PPVs were presented for cancer overall and by cancer site.

Age and sex-specific PPVs with 95% confidence intervals (CIs) were then estimated for any cancer diagnosis, using 10-year age bands (DIAGT Stata module) [42]. Age and sex-specific risk ratios were calculated, comparing observed cancer rates to expected rates in the UK population using UK office for national statistic (ONS) cancer incidence and population size estimates from mid-2011 [43]. The excess cancer risk (observed risk minus expected risk) in symptomatic patients compared to the general population was also estimated.

### Tested-symptomatic cohort

Baseline characteristics of "tested" symptomatic patients in this combined cohort were compared with "untested" symptomatic patients from the initial cohort. Age and sex-specific PPVs for any cancer site were calculated for patients with (a) any concurrent blood test request; (b) any abnormal blood test result; and (c) all normal blood results. The predictive value of each individual blood test abnormality for cancer overall was then examined by age and sex using diagnostic accuracy statistics: We focused on PPVs as they are currently used by NICE to determine criteria for recommending urgent cancer assessment, using a PPV threshold of over 3% [44], and therefore, have most clinical relevance for translation of the findings. Additionally, most of the relevant evidence also focuses on PPVs. Negative predictive value (NPV), sensitivity, specificity, and positive and negative likelihood ratios were also estimated, alongside risk ratios comparing cancer risk in patients with a normal versus abnormal test result using Poisson regression (the risk ratios component was a post hoc analysis). Cancer site-specific PPVs were then calculated for each blood test (post hoc analysis) and used to rank cancer sites in order of likelihood to guide subsequent investigation/referral strategies.

To estimate the potential impact of incorporating information from blood test results into cancer referral recommendations, the baseline number of patients where an urgent referral would be recommended based on their presenting symptom, age and sex was modelled and compared to a model additionally using information from blood test results (post hoc analysis). The numbers of cancer patients detected via urgent referral from each model were compared, assuming 100% guideline adherence with a 3% referral threshold, to estimate the number of additional cancers that could be potentially detected via the urgent referral route or missed.

Sensitivity analyses were conducted using chi-squared tests to compare patient demographics and 1 year cancer incidence in patients identified from the 2 scenarios (blood test followed by symptom versus symptom followed by blood test) to ensure the populations and outcomes were similar.

## Results

### Symptomatic cohort description

A total of 425,549 patients with a new episode of abdominal pain and 52,321 patients with new abdominal bloating were included in the study. Abdominal bloating patients were slightly

**Table 1. Baseline characteristics of patients presenting to their GP with new onset abdominal pain or bloating.**

| | | Abdominal pain patients | | | | Abdominal bloating patients | | |
|---|---|---|---|---|---|---|---|---|
| | Overall N = 425,549 | Tested N = 273,155 | Untested N = 152,394 | P-value | Overall N = 52,321 | Tested N = 36,633 | Untested N = 15,688 | P-value |
| Male sex | 156,590 (37%) | 98,687 (36%) | 57,903 (38%) | <0.001 | 14,255 (27%) | 9,667 (26%) | 4,588 (29%) | <0.001 |
| Age at presentation (years) | | | | <0.001 | | | | <0.001 |
| 30–39 | 95,019 (22%) | 47,045 (17%) | 45,855 (30%) | | 8,820 (17%) | 5,014 (14%) | 3,926 (25%) | |
| 40–49 | 99,406 (23%) | 58,128 (21%) | 41,318 (27%) | | 12,988 (25%) | 8,658 (24%) | 4,284 (27%) | |
| 50–59 | 81,809 (19%) | 54,790 (20%) | 27,724 (18%) | | 11,091 (21%) | 7,956 (22%) | 3,092 (20%) | |
| 60–69 | 71,486 (17%) | 52,386 (19%) | 19,430 (13%) | | 9,188 (18%) | 6,963 (19%) | 2,217 (14%) | |
| 70–79 | 48,656 (11%) | 38,364 (14%) | 10,796 (7%) | | 6,344 (12%) | 5,051 (14%) | 1,296 (8%) | |
| 80 and over | 29,173 (7%) | 22,442 (8%) | 7,271 (5%) | | 3,890 (7%) | 2,991 (8%) | 873 (6%) | |
| Mean (SD, range) | 54 (15.6, 30–104) | 56 (15.6, 30–104) | 50 (14.8, 30–102) | | 55 (15.1, 30–101) | 57 (15.0, 30–101) | 52 (14.9, 30–100) | |
| Median (IQR) | 52 (41–65) | 55 (43–68) | 47 (38–59) | | 53 (43–66) | 55 (45–68) | 49 (40–62) | |
| Year of Presentation[Ψ] | | | | <0.001 | | | | <0.001 |
| 2007–2008 | 122,545 (29%) | 69,713 (26%) | 42,698 (28%) | | 10,975 (21%) | 6,522 (18%) | 3,922 (25%) | |
| 2009–2010 | 107,728 (25%) | 66,783 (24%) | 38,065 (25%) | | 12,563 (24%) | 8,279 (23%) | 4,108 (26%) | |
| 2011–2012 | 89,419 (21%) | 60,097 (22%) | 32,349 (21%) | | 12,515 (24%) | 9,144 (25%) | 3,514 (22%) | |
| 2013–2014 | 68,166 (16%) | 48,589 (18%) | 24,948 (16%) | | 10,375 (20%) | 8,038 (22%) | 2,654 (17%) | |
| 2015–2016 | 37,691 (9%) | 27,973 (10%) | 14,334 (9%) | | 5,893 (11%) | 4,650 (13%) | 1,490 (10%) | |
| IMD quintile | | | | 0.002 | | | | 0.06 |
| 1 (least deprived) | 99,823 (23%) | 64,056 (23%) | 35,767 (23%) | | 12,582 (24%) | 8,878 (24%) | 3,704 (24%) | |
| 2 | 93,261 (22%) | 59,952 (22%) | 33,309 (22%) | | 11,525 (22%) | 8,110 (22%) | 3,415 (22%) | |
| 3 | 89,277 (21%) | 57,617 (21%) | 31,660 (21%) | | 10,817 (21%) | 7,572 (21%) | 3,245 (21%) | |
| 4 | 76,064 (18%) | 48,668 (18%) | 27,396 (18%) | | 9,184 (18%) | 6,397 (17%) | 2,787 (18%) | |
| 5 (most deprived) | 66,851 (16%) | 42,711 (16%) | 24,140 (16%) | | 8,177 (16%) | 5,657 (15%) | 2,520 (16%) | |
| Missing | 273 (0.1%) | 151 (0.1%) | 122 (0.1%) | | 36 (0.1%) | 19 (0.1%) | 17 (0.1%) | |
| Cancer within 12 months | 9,427 (2.2%) | 7,335 (2.7%) | 2,282 (1.5%) | <0.001 | 1,148 (2.2%) | 918 (2.5%) | 245 (1.6%) | <0.001 |
| Number of blood tests* | - | | - | | - | | - | |
| 1–4 | - | 56,662 (21%) | - | | - | 6,849 (19%) | - | |
| 5–8 | - | 69,168 (25%) | - | | - | 8,490 (23%) | - | |
| 9–12 | - | 126,500 (46%) | - | | - | 17,876 (49%) | - | |
| ≥13 | - | 20,825 (8%) | - | | - | 3,418 (9%) | - | |
| Mean (SD, range) | - | 8.3 (4.5, 1–114) | - | | - | 8.6 (4.4, 1–70) | - | |
| Median (IQR) | - | 9 (5–10) | - | | - | 9 (6–11) | - | |

Columns show all patients and those with and without a blood test in the 3 months pre/post presentation; P-value compares tested and untested symptomatic cohorts using chi squared test, SD, standard deviation; IQR, interquartile range; IMD, index of multiple deprivation.

* For 16 prespecified blood tests in the 3 months before the index date.

[Ψ] The progressive decrease of our analysis sample over time reflects the progressive decrease of patients included in CPRD Gold (as it was progressively replaced by CPRD Aurum during our study period, with practices moving out of the Vision IT system which supported CPRD Gold), and the selection of the first eligible symptom presentation in the study period.

older than the abdominal pain group ($p < 0.001$), with a lower proportion aged 30 to 39 years (17% versus 22%), and were more likely to be female (73% versus 63%, $p < 0.001$) (Table 1).

## Predictive value of abdominal symptoms for cancer

Among the analysis sample (patients aged 30 to 104 years), the overall predictive value of abdominal pain or bloating for any cancer in the following 12 months was 2.2% (9,427

**Table 2. PPVs of abdominal symptoms and blood tests for cancer by age and sex.**

| Age group (years) | PPV (95%CI) for cancer in the next 12 months | | | |
|---|---|---|---|---|
| | Abdo pain | Abdo pain + blood request | Abdo pain + any abnormal blood result | Abdo pain + all normal blood results |
| **Men n =** | 156,590 | 98,687 | 56,897 | 41,790 |
| Overall | 2.8 (2.7–2.9) | 3.6 (3.5–3.7) | 4.8 (4.7–5.0) | 1.9 (1.7–2.0) |
| 30–39 | 0.3 (0.2–0.3) | 0.4 (0.3–0.5) | 0.5 (0.3–0.7) | 0.3 (0.2–0.5) |
| 40–49 | 0.6 (0.5–0.7) | 0.8 (0.7–0.9) | 1.1 (0.9–1.3) | 0.4 (0.3–0.6) |
| 50–59 | 1.9 (1.8–2.1) | 2.3 (2.1–2.5) | 3.1 (2.8–3.4) | 1.4 (1.1–1.6) |
| 60–69 | 4.2 (4.0–4.5) | 4.7 (4.4–5.0) | 6.3 (5.9–6.7) | 2.6 (2.3–2.9) |
| 70–79 | 6.9 (6.5–7.2) | 7.3 (6.8–7.7) | 8.7 (8.2–9.3) | 4.6 (4.0–5.1) |
| ≥80 | 8.6 (8.0–9.1) | 8.9 (8.3–9.6) | 10.2 (9.4–11.0) | 5.2 (4.2–6.2) |
| **Women n =** | 268,959 | 174,468 | 86,208 | 88,260 |
| Overall | 1.9 (1.8–1.9) | 2.2 (2.1–2.3) | 3.1 (3.0–3.3) | 1.2 (1.2–1.3) |
| 30–39 | 0.4 (0.4–0.5) | 0.4 (0.4–0.5) | 0.5 (0.4–0.6) | 0.4 (0.3–0.5) |
| 40–49 | 0.8 (0.7–0.8) | 0.8 (0.7–0.9) | 1.2 (1.0–1.3) | 0.5 (0.4–0.7) |
| 50–59 | 1.5 (1.4–1.6) | 1.6 (1.4–1.7) | 2.2 (2.0–2.4) | 1.0 (0.8–1.1) |
| 60–69 | 3.1 (2.9–3.3) | 3.4 (3.2–3.6) | 4.9 (4.5–5.2) | 1.9 (1.7–2.1) |
| 70–79 | 4.4 (4.1–4.6) | 4.4 (4.2–4.7) | 5.9 (5.5–6.3) | 2.7 (2.4–3.0) |
| ≥80 | 5.1 (4.8–5.5) | 5.2 (4.8–5.6) | 6.5 (6.0–7.0) | 3.2 (2.7–3.7) |
| Age group (years) | PPV (95% CI) for cancer in the next 12 months | | | |
| | Abdo bloating | Abdo bloating + blood request | Abdo bloating + any abnormal blood result | Abdo bloating + all normal blood results |
| **Men n =** | 14,255 | 9,667 | 5,600 | 4,067 |
| Overall | 2.6 (2.3–2.8) | 3.2 (2.8–3.5) | 4.1 (3.6–4.7) | 1.8 (1.4–2.3) |
| 30–39 | 0.3 (0.1–0.7) | 0.5 (0.2–1.2) | 0.6 (0.1–1.9) | 0.4 (0.0–1.4) |
| 40–49 | 0.5 (0.3–0.8) | 0.7 (0.4–1.2) | 0.9 (0.4–1.7) | 0.5 (0.1–1.2) |
| 50–59 | 1.6 (1.2–2.2) | 1.9 (1.4–2.6) | 2.5 (1.7–3.5) | 1.3 (0.7–2.2) |
| 60–69 | 3.0 (2.5–3.7) | 3.3 (2.6–4.1) | 3.9 (2.9–5.1) | 2.4 (1.6–3.6) |
| 70–79 | 6.3 (5.3–7.5) | 7.0 (5.8–8.4) | 9.1 (7.4–11.0) | 3.4 (2.1–5.2) |
| ≥80 | 6.7 (5.3–8.4) | 7.1 (5.4–9.1) | 7.5 (5.5–9.9) | 5.9 (3.0–10.3) |
| **Women n =** | 38,066 | 26,966 | 12,291 | 14,675 |
| Overall | 2.1 (1.9–2.2) | 2.3 (2.1–2.5) | 3.5 (3.2–3.9) | 1.2 (1.1–1.4) |
| 30–39 | 0.4 (0.3–0.6) | 0.4 (0.3–0.7) | 0.7 (0.3–1.2) | 0.3 (0.1–0.6) |
| 40–49 | 0.7 (0.6–0.9) | 0.9 (0.7–1.1) | 1.4 (1.0–1.9) | 0.5 (0.3–0.7) |
| 50–59 | 1.6 (1.3–1.9) | 1.7 (1.4–2.0) | 2.7 (2.1–3.3) | 0.9 (0.6–1.3) |
| 60–69 | 3.5 (3.1–4.0) | 3.8 (3.3–4.4) | 5.3 (4.4–6.3) | 2.4 (1.9–3.1) |
| 70–79 | 4.7 (4.1–5.4) | 4.4 (3.8–5.2) | 6.2 (5.1–7.4) | 2.7 (2.0–3.6) |
| ≥80 | 4.6 (3.9–5.5) | 5.0 (4.1–6.0) | 6.7 (5.4–8.2) | 2.5 (1.6–3.8) |

Red shading represents a PPV of 10% or higher; orange shading represent a PPV of 3% or higher, the threshold above which the National Institute for Health and Care Excellence recommends investigation for cancer; yellow shading represents a PPV of 2% to 3%.

CI, confidence interval; PPV, positive predictive value.

abdominal pain patients and 1,148 abdominal bloating patients) (Table 1). Cancer incidence was higher in males with either abdominal symptom than in females (PPV 2.8% and 2.6% in males compared to 1.9% and 2.1% in females, respectively, for abdominal pain and bloating) (Table 2). Estimation of age-specific PPVs for cancer using 10-year age bands were between 0.3% and 8.6% for abdominal pain and 0.3% to 6.7% for abdominal bloating. For either symptom, patients of both sexes aged 60 years and over had PPVs above the 3% NICE threshold for referral (PPVs in patients ≥60 years with abdominal pain 3.1% to 8.6% and abdominal

bloating 3.0% to 6.7%, Table 2). Patients seen in primary care for either symptom had more than double the risk of cancer compared to the general population (age and sex-specific risk ratios 2.5 to 4.0 for abdominal pain and 2.1 to 4.7 for abdominal bloating), with an excess risk observed in all age groups except males aged 30 to 39 with abdominal bloating. The excess risk was concentrated in patients aged 60 years and over, increasing up to 5% (Table D in S1 Supplementary File).

Among patients of both sexes with either abdominal symptom, PPVs for individual cancer sites were all <1%. The most common cancer sites in males were colon (PPV 0.6% and 0.5%), prostate (PPV 0.4% and 0.3%), and pancreas (PPV 0.3%); and in females were colon (PPV 0.4% and 0.2%), breast (PPV 0.2% and 0.3%), ovary (PPV 0.2% and 0.6%), and pancreas (PPV 0.2% and 0.1%) (respectively for abdominal pain and abdominal bloating, Table E in S1 Supplementary File).

Among patients diagnosed with cancer, the distribution of cancer by site in males was similar in both abdominal pain and abdominal bloating cohorts, apart from liver cancer which accounted for twice as many cancers in abdominal bloating patients (6.3% versus 3.3% of cancers, $p$ = 0.002). In females, ovarian cancer accounted for 1 in 3 of all cancers diagnosed in abdominal bloating patients, compared to 11% of cancers in abdominal pain patients ($p$ < 0.001) (Table F in S1 Supplementary File).

## Tested symptomatic cohort description

From the symptomatic cohorts described above, 273,155/425,549 abdominal pain patients and 36,633/52,321 abdominal bloating patients were identified who had 1 or more specified blood tests within 3 months of symptomatic presentation (Tables 1 and G in S1 Supplementary File). When compared with non-tested patients, symptomatic tested patients were on average older, with a median age of 55 years compared to 47 and 49 years in untested patients with abdominal pain and bloating, respectively (Table 1). Sensitivity analysis comparing tested symptomatic patients identified from the 2 scenarios (blood test before symptom versus blood test after symptom) showed that the populations were similar in size, broadly comparable with regards to demographics and had similar 1 year cancer risk (Tables H and I in S1 Supplementary File).

## Predictive value of blood tests in patients with abdominal symptoms for cancer

Cancer was more likely in tested than untested symptomatic patients, with a PPV for any cancer of 2.7% or 2.5% in tested abdominal pain and bloating patients, respectively, compared to 1.5% or 1.6% ($p$ < 0.001) (Table 1), and higher PPVs in tested males (3.6% or 3.2%) than females (2.2% or 2.3%) (Table 2). Age-specific PPVs for cancer were higher in tested (versus all) symptomatic patients for each age group, but the threshold where they reached >3% remained at ≥60 years (Table 2).

For both symptoms, patients with any abnormal blood test results had elevated cancer risk when compared with all tested patients, and those with all-normal blood test results had lower cancer risk (Table 2). Symptomatic patients with all-normal blood results did not have a risk of cancer above 3% until age 70 or older, with the age threshold depending on sex and presenting symptom.

The individual blood test results most strongly predictive of cancer in symptomatic patients were low albumin, high platelets, high PSA, and high CA125 (Fig 2 and Tables J and K in S1 Supplementary File). All blood test abnormalities were associated with cancer risk >2% in patients with these symptoms, except for low WBC in females with either symptom, low ferritin in females with bloating, and raised ALT in men with bloating (Fig 2). Normal results in no

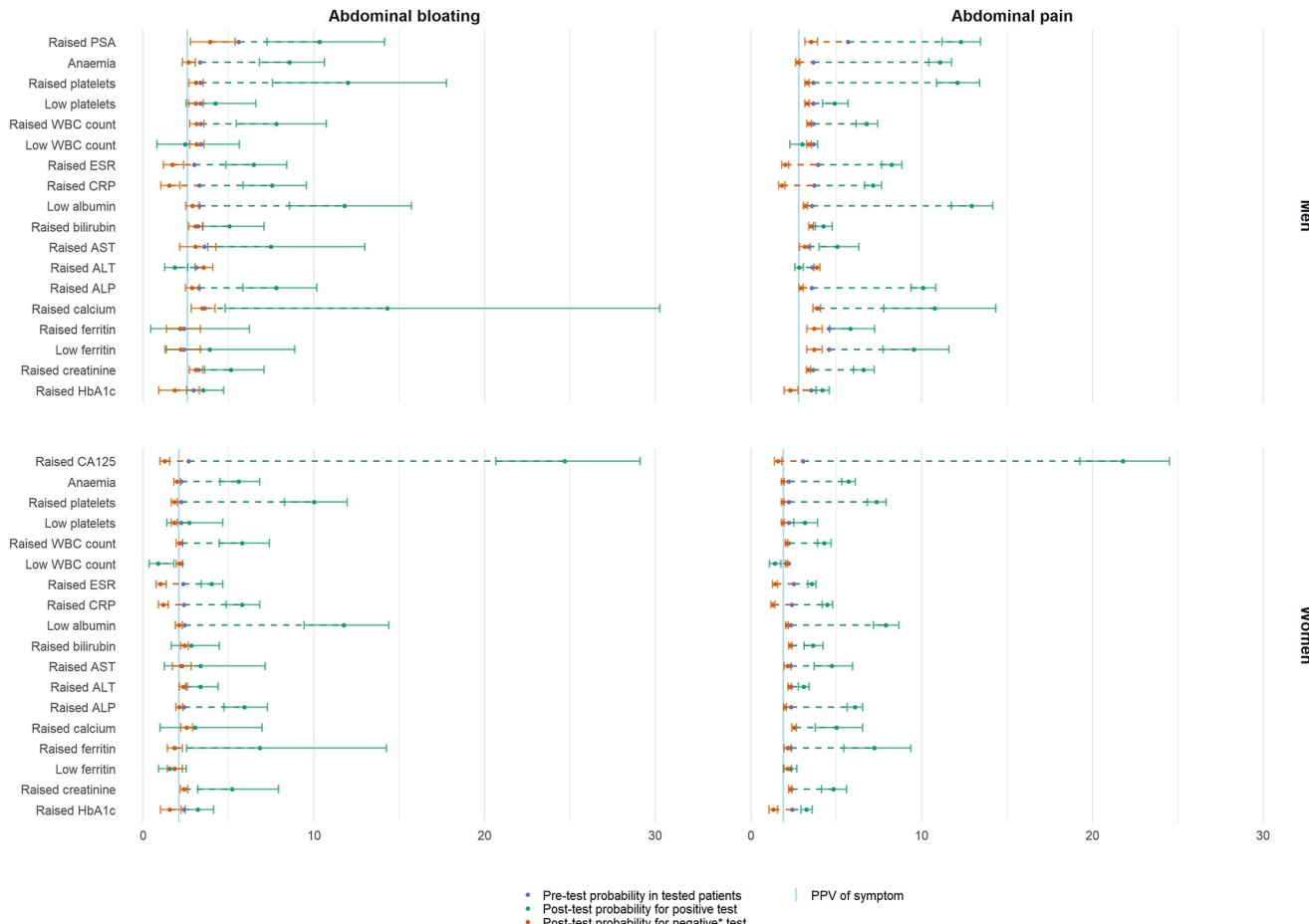

**Fig 2. Pretest and posttest PPVs for cancer in 12 months by sex in patients with abdominal pain/bloating using results of commonly used blood tests.**
Blue line represents the PPV for the symptom alone, blue dot represents the PPV of having a blood test done, orange dot represents the PPV of an abnormal result and purple dot represents the PPV of a normal result. Error bars and solid lines represent 95% CIs. CI, confidence interval; PPV, positive predictive value.

single blood test were sufficient to be used as a safe "rule-out" test for cancer alone, as reflected by the negative likelihood ratios >0.2 [39,45,46]. The cancer sites with the highest PPVs for each blood test are shown in Tables 3–6 and L in S1 Supplementary File. Risk ratios comparing 1 year cancer risk in symptomatic patients with a normal versus abnormal blood test result are shown in Tables M and N in S1 Supplementary File for each of the 19 blood test abnormalities examined.

Considering age-specific PPVs, in male and female patients with either symptom aged ≥60 almost all individual blood test abnormalities had a PPV for cancer of >3% (Tables 3–6). The number of blood test abnormalities with PPVs above the 3% threshold progressively dropped in the 50 to 59 and 40 to 49 age groups, but many remained >3% or between 2% and 3%, especially in males. In symptomatic patients aged 30 to 39, the PPVs for cancer associated with blood test abnormalities were lower; the only symptom/result combinations with PPVs >3% were for males with abdominal pain and either anaemia, low albumin, or low ferritin and females with either symptom and raised CA125 (Tables 3–6).

Comparison of models basing urgent cancer referral recommendations on information from presenting symptom-age-sex versus additionally considering information from blood

**Table 3. PPVs of blood test abnormalities for cancer within 12 months in male patients with abdominal pain by age.**

| Blood abnormality | PPV in males with abdominal pain (95% CI) by age group (years) | | | | | | Cancer sites from highest to lowest risk |
| --- | --- | --- | --- | --- | --- | --- | --- |
| | 30–39 | 40–49 | 50–59 | 60–69 | 70–79 | ≥80 | |
| **Baseline risk** | 0.4 (0.3–0.5) | 0.8 (0.7–0.9) | 2.3 (2.1–2.5) | 4.7 (4.4–5.0) | 7.3 (6.8–7.7) | 8.9 (8.3–9.6) | |
| Raised platelets | 2.4 (0.9–5.2) | 4.9 (3.1–7.3) | 11.4 (8.9–14.2) | 16.3 (13.5–19.4) | 18.0 (14.6–21.7) | 15.0 (10.9–19.9) | **Bowel, lung, gastro-oesophageal,** lymphoma, prostate, renal tract, pancreas, CUP, other, liver |
| Raised calcium | 2.9 (0.3–9.9) | 4.6 (1.0–12.9) | 10.0 (3.8–20.5) | 17.1 (9.7–27.0) | 16.4 (7.8–28.8) | 15.0 (5.7–29.8) | **Lung, bowel, CUP, prostate,** liver, myeloma, lymphoma, pancreas, renal tract, other, gastro-oesophageal, leukaemia |
| Anaemia | 4.0 (2.1–6.7) | 5.1 (3.6–7.1) | 8.9 (7.2–10.8) | 11.8 (10.3–13.4) | 12.8 (11.5–14.3) | 12.4 (11.2–13.8) | **Bowel, gastro-oesophageal,** prostate, renal tract, lung, CUP, pancreas, lymphoma, other, liver, leukaemia, myeloma |
| Low albumin | 3.8 (1.1–9.6) | 1.8 (0.6–4.3) | 8.9 (6.3–12.0) | 14.5 (11.9–17.3) | 16.0 (13.4–18.9) | 16.0 (13.5–18.8) | **Bowel, lung, gastro-oesophageal, CUP, liver,** prostate, pancreas, lymphoma, other, renal tract, myeloma, head & neck, brain and CNS |
| Raised ALP | 1.8 (0.9–3.1) | 3.3 (2.3–4.6) | 6.3 (5.1–7.8) | 13.2 (11.6–15.0) | 15.9 (14.0–18.0) | 15.3 (13.0–17.8) | **Bowel, pancreas, prostate, CUP, liver, lung,** gastro-oesophageal, other, renal tract, lymphoma |
| Raised AST | 0.4 (0.0–2.4) | 1.1 (0.3–2.8) | 2.9 (1.4–5.2) | 10.0 (6.7–14.3) | 10.6 (6.2–16.6) | 18.3 (10.6–28.4) | Pancreas, liver, gastro-oesophageal, bowel, other, prostate, lung, CUP, renal tract, lymphoma |
| Low ferritin | 3.6 (0.4–12.3) | 2.8 (0.6–8.0) | 6.8 (3.7–11.4) | 8.4 (5.0–13.1) | 10.9 (7.1–15.7) | 18.5 (12.9–25.2) | **Bowel, gastro-oesophageal,** other, prostate, renal tract, pancreas, CUP, lymphoma, lung, liver, myeloma |
| Raised CRP | 1.0 (0.5–1.6) | 2.4 (1.8–3.1) | 5.2 (4.3–6.2) | 9.5 (8.3–10.7) | 12.4 (10.8–14.0) | 15.5 (13.3–17.9) | **Bowel,** lung, prostate, pancreas, lymphoma, CUP, gastro-oesophageal, renal tract, liver, other |
| Raised ESR | 1.3 (0.6–2.4) | 2.9 (2.1–4.0) | 5.0 (4.0–6.1) | 9.9 (8.6–11.3) | 12.7 (11.2–14.4) | 15.0 (12.8–17.4) | **Bowel,** lung, gastro-oesophageal, pancreas, prostate, CUP, lymphoma, liver, renal tract, other, leukaemia |
| Raised total WBC | 1.2 (0.6–2.3) | 1.9 (1.2–2.8) | 5.8 (4.6–7.2) | 8.8 (7.3–10.5) | 11.4 (9.4–13.5) | 13.5 (10.9–16.4) | **Bowel,** lung, gastro-oesophageal, pancreas, CUP, prostate, leukaemia, lymphoma, other, renal tract, liver |
| Raised ALT | 0.3 (0.1–0.5) | 0.7 (0.5–1.0) | 2.0 (1.6–2.4) | 5.5 (4.7–6.3) | 8.9 (7.4–10.5) | 16.0 (12.7–19.8) | Pancreas, liver, bowel, prostate, gastro-oesophageal, CUP, lung, other, renal tract, lymphoma |
| Raised ferritin | 1.7 (0.2–6.1) | 2.4 (0.9–5.1) | 3.5 (1.8–6.2) | 6.6 (4.0–10.3) | 9.7 (5.9–14.9) | 14.2 (8.8–21.3) | CUP, lung, prostate, pancreatic, renal tract, myeloma, bowel, lymphoma, gastro-oesophageal, other, leukaemia, liver |
| Low total WBC | 0.3 (0.0–1.7) | 0.4 (0.1–1.5) | 1.8 (0.7–3.7) | 5.2 (2.9–8.4) | 6.7 (3.7–11.0) | 13.0 (7.6–20.3) | Prostate, bowel, lymphoma, pancreas, leukaemia, myeloma, gastro-oesophageal, liver |
| Low platelets | 0.3 (0.0–1.9) | 2.1 (1.1–3.7) | 2.8 (1.6–4.5) | 5.5 (3.9–7.5) | 7.1 (5.4–9.2) | 8.7 (6.4–11.5) | Pancreas, prostate, liver, lymphoma, leukaemia, gastro-oesophageal, lung, renal tract, bowel, other haem, other, CUP, myeloma |
| Raised bilirubin | 0.6 (0.2–1.3) | 1.1 (0.6–1.8) | 2.5 (1.6–3.3) | 6.2 (5.0–7.6) | 8.3 (6.7–10.1) | 9.4 (7.0–12.1) | Pancreas, liver, bowel, prostate, gastro-oesophageal, CUP, other, lung, renal tract, lymphoma |
| Raised creatinine | 1.1 (0.1–4.1) | 0.3 (0.0–1.5) | 2.9 (1.8–4.6) | 5.7 (4.4–7.1) | 7.7 (6.6–8.9) | 8.5 (7.4–9.7) | **Bowel,** prostate, renal tract, lung, gastro-oesophageal, CUP, lymphoma, liver, other, pancreas, leukaemia, myeloma |
| Raised HbA1c | 0.0 (0.0–0.9) | 0.5 (0.2–1.0) | 2.5 (1.9–3.3) | 4.7 (4.0–5.5) | 6.0 (5.1–7.0) | 8.2 (6.6–10.0) | **Bowel,** pancreas, prostate, gastro-oesophageal, lung, CUP, renal tract, lymphoma, other, liver |
| Raised PSA | 0.0 (0.0–30.8) | 7.7 (2.1–18.5) | 10.2 (7.1–14.0) | 12.1 (10.2–14.3) | 13.1 (11.2–15.1) | 12.7 (10.4–15.3) | **Prostate, bowel,** lung, renal tract, pancreas, gastro-oesophageal, CUP, lymphoma, other, liver, myeloma |

Red = PPV of ≥10%; orange = PPV of ≥3% (NICE threshold for recommending cancer investigation); yellow = PPV ≥2%. Cancer sites in bold have PPVs >1%; all listed cancer sites have PPVs >0.1%; CUP, cancer of unknown primary; other haem = other haematological cancer, CI, confidence interval; CRP, c reactive protein; ESR, erythrocyte sedimentation rate; WBC, white blood cell count; PSA, prostate specific antigen.

test results indicated that for every 1,000 patients with abdominal bloating who had blood tests: 74 additional urgent referrals could be generated and 10 referrals could be avoided. For abdominal pain the respective numbers were up to 68 additional referrals and 2 referrals avoided. Among the 1,000 patients, all those with cancer referrable using the model solely using symptom-age-sex information were also referred using the posttest model, but 5 additional cancer patients were identified for urgent referral using blood tests (3 abdominal bloating patients and 2 abdominal pain) (Box 1 in Fig 3).

**Table 4. PPVs of blood test abnormalities for cancer within 12 months in female patients with abdominal pain by age.**

| Blood abnormality | PPV in females with abdominal pain (95% CI) by age group (years) | | | | | | Cancer sites from highest to lowest risk |
|---|---|---|---|---|---|---|---|
| | 30–39 | 40–49 | 50–59 | 60–69 | 70–79 | ≥80 | |
| **Baseline risk** | 0.4 (0.4–0.5) | 0.8 (0.7–0.9) | 1.6 (1.4–1.7) | 3.4 (3.2–3.6) | 4.4 (4.2–4.7) | 5.2 (4.8–5.6) | |
| Raised platelets | 0.7 (0.3–1.3) | 2.6 (1.9–3.4) | 5.4 (4.3–6.6) | 12.6 (10.9–14.3) | 13.3 (11.5–15.2) | 13.3 (11.3–15.5) | **Bowel, ovary,** CUP, lung, gastro-oesophageal, pancreas, breast, other, renal tract, uterus, lymphoma, cervix, liver |
| Raised calcium | 0.0 (0.0–7.7) | 0.0 (0.0–3.7) | 3.9 (1.7–7.6) | 4.6 (2.4–7.9) | 7.4 (4.3–11.5) | 7.8 (4.3–12.8) | CUP, bowel, lung, lymphoma, renal tract, pancreas, myeloma, gastro-oesophageal, ovary, liver, breast |
| Anaemia | 0.4 (0.2–0.7) | 2.0 (1.6–2.6) | 4.8 (3.7–6.1) | 12.1 (10.5–14.0) | 11.0 (9.6–12.5) | 9.9 (8.7–11.1) | **Bowel,** ovary, CUP, lung, pancreas, gastro-oesophageal, lymphoma, breast, other, renal tract, myeloma, uterus, cervix |
| Low albumin | 1.0 (0.4–2.0) | 2.5 (1.4–3.9) | 5.2 (3.7–7.1) | 11.1 (9.0–13.6) | 12.2 (10.2–14.5) | 10.7 (9.1–12.5) | **Bowel, ovary,** CUP, lung, lymphoma, pancreas, gastro-oesophageal, renal tract, other, breast, liver, myeloma, cervix |
| High ALP | 0.6 (0.2–1.2) | 1.8 (1.2–2.6) | 4.2 (3.4–5.1) | 8.2 (7.2–9.4) | 9.0 (7.7–10.3) | 9.7 (8.3–11.3) | **Bowel, pancreas,** CUP, lung, ovary, liver, breast, other, gastro-oesophageal, lymphoma, renal tract, uterus |
| High AST | 0.0 (0.0–2.1) | 1.7 (0.6–4.0) | 3.7 (2.0–6.2) | 7.6 (5.0–11.2) | 9.2 (5.6–14.0) | 6.8 (2.8–13.5) | **Pancreas,** bowel, liver, lung, ovary, other, gastro-oesophageal, CUP, lymphoma, breast |
| Low ferritin | 0.3 (0.1–0.7) | 1.1 (0.7–1.6) | 2.3 (1.4–3.7) | 6.2 (4.0–9.0) | 7.5 (5.0–10.8) | 10.5 (7.3–14.5) | **Bowel,** breast, gastro-oesophageal, cervix, pancreas |
| Raised CRP | 0.9 (0.6–1.3) | 1.3 (0.9–1.7) | 3.3 (2.8–3.9) | 7.2 (6.4–8.1) | 8.2 (7.2–9.2) | 8.7 (7.5–10.1) | **Bowel,** ovary, pancreas, CUP, lung, breast, gastro-oesophageal, lymphoma, other, renal tract, cervix, uterus, liver |
| Raised ESR | 0.5 (0.3–0.8) | 1.1 (0.9–1.5) | 2.4 (2.0–2.8) | 5.1 (4.6–5.8) | 6.3 (5.6–7.1) | 7.0 (6.0–8.1) | Bowel, ovarian, pancreas, CUP, lung, breast, lymphoma, gastro-oesophageal, renal tract, other |
| Raised total WBC | 0.7 (0.4–1.1) | 1.7 (1.2–2.4) | 3.3 (2.5–4.4) | 8.1 (6.8–9.6) | 8.4 (7.0–10.1) | 9.1 (7.5–10.9) | **Bowel,** lung, CUP, ovary, pancreas, breast, other, gastro-oesophageal, lymphoma, cervix, leukaemia, renal tract |
| High ALT | 0.6 (0.3–1.0) | 0.9 (0.6–1.4) | 2.4 (1.9–2.9) | 4.8 (4.0–5.7) | 6.4 (5.2–7.9) | 8.8 (6.7–11.3) | Pancreas, bowel, CUP, breast, lung, ovary, other, liver, lymphoma, gastro-oesophageal |
| Raised ferritin | 2.6 (0.1–13.5) | 1.6 (0.0–8.7) | 4.5 (1.7–9.6) | 8.7 (4.9–13.9) | 7.8 (4.2–13.0) | 10.7 (6.3–16.9) | **Bowel, pancreas,** CUP, lung, other, breast, ovary, gastro-oesophageal, lymphoma, uterus, cervix, sarcoma, renal tract, myeloma |
| Low total WBC | 0.3 (0.1–1.0) | 0.8 (0.4–1.6) | 1.2 (0.6–2.0) | 1.8 (1.0–2.9) | 3.8 (2.3–6.0) | 2.7 (1.2–5.3) | Breast, lymphoma, bowel, ovary, leukaemia |
| Low platelets | 0.9 (0.3–2.1) | 1.7 (0.7–3.3) | 1.2 (0.4–2.9) | 5.8 (3.7–8.5) | 6.4 (4.3–9.2) | 3.9 (2.2–6.5) | Pancreas, lymphoma, myeloma, leukaemia, liver, breast, CUP, cervix, ovary, other |
| Raised bilirubin | 0.1 (0.0–0.6) | 1.4 (0.8–2.4) | 3.0 (1.9–4.4) | 5.4 (3.9–7.3) | 6.3 (4.6–8.3) | 9.2 (6.7–12.2) | **Pancreas,** liver, other, bowel, CUP, breast, ovary, gastro-oesophageal, renal tract |
| Raised creatinine | 0.0 (0.0–5.8) | 2.7 (0.6–7.6) | 2.4 (0.6–5.9) | 4.0 (2.3–6.3) | 5.4 (4.0–7.0) | 5.3 (4.3–6.5) | Bowel, CUP, renal tract, lymphoma, lung, ovary, breast, gastro-oesophageal, pancreas, other, liver |
| Raised HbA1c | 0.6 (0.2–1.6) | 0.9 (0.5–1.5) | 1.8 (1.3–2.5) | 3.8 (3.1–4.5) | 4.6 (3.8–5.4) | 5.4 (4.3–6.7) | Bowel, pancreas, breast, ovary, lung, CUP, gastro-oesophageal, renal tract, other, lymphoma |
| Raised CA125 | 4.2 (1.4–9.5) | 9.2 (6.4–12.6) | 22.5 (16.6–29.3) | 44.8 (35.9–54.0) | 46.0 (37.1–55.1) | 33.9 (22.3–47.0) | **Ovary, bowel, pancreas, CUP,** uterus, other, sarcoma, lung, gastro-oesophageal, lymphoma, renal tract, liver, breast, melanoma, myeloma, cervix |

Red = PPV of ≥10%; orange = PPV of ≥3% (NICE threshold for recommending cancer investigation); yellow = PPV ≥2%. Cancer sites in bold have PPVs >1%; all listed cancer sites have PPVs >0.1%; CUP, cancer of unknown primary; other haem = other haematological cancer, CI, confidence interval; CRP, c reactive protein; ESR, erythrocyte sedimentation rate; WBC, white blood cell count; PSA, prostate specific antigen.

## Discussion

We present the risk of cancer (overall and by site) for 19 abnormal blood test results in patients presenting to primary care with abdominal pain or bloating, by age and sex. In male and female patients aged 60 years and over presenting to primary care with abdominal pain or

**Table 5. PPVs of blood test abnormalities for cancer within 12 months in male patients with abdominal bloating by age.**

| Blood abnormality | PPV in males with abdominal bloating (95% CI) by age group (years) | | | | | | Cancer sites from highest to lowest risk |
|---|---|---|---|---|---|---|---|
| | 30–39 | 40–49 | 50–59 | 60–69 | 70–79 | ≥80 | |
| **Baseline risk** | 0.5 (0.2–1.2) | 0.7 (0.4–1.2) | 1.9 (1.4–2.6) | 3.3 (2.6–4.1) | 7.0 (5.8–8.4) | 7.1 (5.4–9.1) | |
| Raised platelets | 0.0 (0.0–30.8) | 6.3 (0.8–20.8) | 9.1 (1.9–24.3) | 15.2 (6.3–28.9) | 13.3 (3.8–30.7) | 20.8 (7.1–42.2) | **Bowel, CUP, lung, gastro-oesophageal, other, lymphoma,** liver, prostate, sarcoma, other haem |
| Raised calcium | 0.0 (0.0–60.2) | 0.0 (0.0–36.9) | 14.3 (0.4–57.9) | 30.0 (6.7–65.2) | 0.0 (0.0–84.2) | 25.0 (0.6–80.6) | **Lung, prostate, CUP, myeloma, head and neck** |
| Anaemia | 0.0 (0.0–14.8) | 6.0 (1.7–14.6) | 6.8 (3.0–13.0) | 8.5 (4.8–13.6) | 10.8 (7.2–15.5) | 8.7 (5.6–12.8) | **Bowel, liver, gastro-oesophageal,** prostate, lymphoma, pancreas, CUP, lung, renal tract, brain and CNS, other haem, melanoma, myeloma, sarcoma, leukaemia |
| Low albumin | 0.0 (0.0–18.5) | 3.6 (0.1–18.3) | 8.6 (2.9–19.0) | 14.3 (7.6–23.6) | 15.1 (7.8–25.4) | 14.1 (7.3–23.8) | **Liver, bowel, pancreas, lung,** gastro-oesophageal, myeloma, lymphoma, CUP, renal tract, other, sarcoma, brain and CNS, testis, other haem |
| Raised ALP | 2.3 (0.1–12.3) | 3.1 (0.6–8.8) | 5.2 (2.1–10.4) | 7.7 (4.0–13.1) | 14.0 (8.5–21.2) | 11.3 (5.3–20.3) | **Liver, pancreas,** prostate, bowel, gastro-oesophageal, CUP, renal tract, brain and CNS, lung, myeloma, lymphoma, leukaemia |
| Raised AST | 0.0 (0.0–21.8) | 2.9 (0.1–14.9) | 4.8 (0.6–16.2) | 7.1 (0.9–23.5) | 15.0 (3.2–37.9) | 42.9 (9.9–81.6) | **Pancreas, prostate,** CUP, bowel, liver, gastro-oesophageal |
| Low ferritin | 0.0 (0.0–41.0) | 0.0 (0.0–21.8) | 0.0 (0.0–12.3) | 3.8 (0.1–19.6) | 5.7 (0.7–19.2) | 11.8 (1.5–36.4) | **Gastro-oesophageal, bowel,** prostate, other |
| Raised CRP | 2.7 (0.3–9.3) | 1.3 (0.2–4.7) | 6.0 (2.9–10.8) | 6.4 (3.5–10.8) | 15.4 (10.2–21.9) | 13.0 (6.9–21.7) | **Bowel,** gastro-oesophageal, pancreas, lymphoma, prostate, CUP, renal tract, lung, liver, sarcoma, testis, other, head and neck, other haem |
| Raised ESR | 2.0 (0.0–10.4) | 2.5 (0.5–7.3) | 5.7 (2.6–10.5) | 5.3 (2.6–9.6) | 10.5 (6.2–16.3) | 9.7 (5.0–16.8) | **Bowel,** prostate, lung, pancreas, liver, gastro-oesophageal, lymphoma, renal tract, CUP, myeloma, other |
| Raised total WBC | 2.6 (0.1–13.8) | 1.3 (0.0–6.9) | 6.1 (2.0–13.7) | 7.6 (3.3–14.5) | 17.4 (10.1–27.1) | 8.5 (2.4–20.4) | **Bowel,** liver, prostate, leukaemia, gastro-oesophageal, CUP, lung, other haem, brain and CNS, renal tract, other, lymphoma |
| Raised ALT | 0.4 (0.0–2.3) | 0.6 (0.1–1.8) | 1.1 (0.4–2.5) | 2.4 (1.0–4.7) | 7.4 (3.5–13.7) | 12.2 (4.1–26.2) | Liver, pancreas, CUP, bowel, gastro-oesophageal, renal tract |
| Raised ferritin | 0.0 (0.0–45.9) | 3.1 (0.1–16.2) | 0.0 (0.0–10.9) | 2.5 (0.1–13.2) | 4.5 (0.1–22.8) | 0.0 (0.0–45.9) | Lymphoma, CUP, pancreas |
| Low total WBC | 0.0 (0.0–12.8) | 0.0 (0.0–6.5) | 5.4 (0.7–18.2) | 5.3 (0.6–17.7) | 3.2 (0.1–16.7) | 0.0 (0.0–20.6) | Liver, pancreas, lymphoma, prostate |
| Low platelets | 0.0 (0.0–13.2) | 3.6 (0.4–12.3) | 1.4 (0.0–7.4) | 5.1 (1.7–11.5) | 8.3 (3.9–15.2) | 1.6 (0.0–8.4) | **Liver,** pancreas, lymphoma, prostate, renal tract, gastro-oesophageal |
| Raised bilirubin | 0.0 (0.0–5.4) | 2.5 (0.5–7.1) | 2.4 (0.5–6.8) | 6.0 (2.8–11.1) | 10.3 (5.5–17.4) | 9.3 (3.1–20.3) | **Liver,** pancreas, prostate, bowel, lung, lymphoma, brain and CNS, renal tract, CUP, head and neck, gastro-oesophageal |
| Raised creatinine | 0.0 (0.0–28.5) | 2.3 (0.1–12.0) | 4.8 (1.0–13.5) | 0.8 (0.0–4.3) | 8.2 (4.9–12.7) | 5.5 (2.8–9.3) | Pancreas, prostate, bowel, renal tract, lymphoma, CUP, gastro-oesophageal, liver, lung, other, head and neck |
| Raised HbA1c | 0.0 (0.0–9.7) | 1.5 (0.2–5.4) | 2.6 (1.1–5.3) | 2.2 (0.9–4.3) | 6.1 (3.6–9.5) | 7.1 (3.1–13.6) | Pancreas, bowel, prostate, renal tract, liver, lung, CUP |
| Raised PSA | – | 0.0 (0.0–84.2) | 7.7 (0.9–25.1) | 9.7 (4.8–17.1) | 8.7 (4.4–15.1) | 15.3 (7.9–25.7) | **Prostate, bowel,** pancreas, gastro-oesophageal, renal tract, CUP, myeloma, lymphoma, other |

Red = PPV of ≥10%; orange = PPV of ≥3% (NICE threshold for recommending cancer investigation); yellow = PPV ≥2%. Cancer sites in bold have PPVs >1%; all listed cancer sites have PPVs >0.1%; CUP, cancer of unknown primary; other haem = other haematological cancer, CI, confidence interval; CRP, c reactive protein; ESR, erythrocyte sedimentation rate; WBC, white blood cell count; PSA, prostate specific antigen.

**Table 6. PPVs of blood test abnormalities for cancer within 12 months in female patients with abdominal bloating by age.**

| | PPV in females with abdominal bloating (95% CI) by age group (years) | | | | | | Cancer sites from highest to lowest risk |
|---|---|---|---|---|---|---|---|
| **Blood abnormality** | 30–39 | 40–49 | 50–59 | 60–69 | 70–79 | ≥80 | |
| **Baseline risk** | **0.4 (0.3–0.7)** | **0.9 (0.7–1.1)** | **1.7 (1.4–2.0)** | **3.8 (3.3–4.4)** | **4.4 (3.8–5.2)** | **5.0 (4.1–6.0)** | |
| Raised platelets | 0.9 (0.0–4.8) | 4.9 (2.6–8.3) | 8.2 (5.0–12.4) | 16.2 (11.3–22.0) | 17.6 (12.0–24.4) | 13.0 (7.6–20.3) | **Ovary, bowel,** CUP, lung, pancreas, breast, gastro-oesophageal, sarcoma, other, uterus |
| Raised calcium | 0.0 (0.0–33.6) | 0.0 (0.0–21.8) | 2.7 (0.1–14.2) | 3.3 (0.1–17.2) | 2.7 (0.1–14.2) | 5.6 (0.7–18.7) | Gastro-oesophageal, lung, CUP, breast, myeloma |
| Anaemia | 0.0 (0.0–1.8) | 1.9 (0.8–3.7) | 3.8 (1.7–7.0) | 8.7 (5.1–13.8) | 9.4 (5.9–14.0) | 10.8 (7.6–14.7) | **Bowel, ovary,** CUP, lung, sarcoma, breast, pancreas, lymphoma, liver, renal tract, gastro-oesophageal, uterus, other haem, other |
| Low albumin | 2.0 (0.0–10.4) | 4.6 (1.3–11.4) | 8.9 (4.5–15.3) | 16.8 (10.7–24.5) | 15.6 (10.0–22.7) | 13.6 (8.7–19.8) | **Ovary, bowel, CUP,** lung, lymphoma, breast, renal tract, other, sarcoma, liver, other haem, gastro-oesophageal, pancreas |
| Raised ALP | 2.1 (0.3–7.5) | 0.9 (0.1–3.3) | 4.2 (2.3–6.9) | 10.2 (7.0–14.2) | 7.1 (4.3–11.0) | 8.1 (4.7–12.8) | **Ovary,** CUP, lung, bowel, liver, pancreas, breast, gastro-oesophageal, lymphoma, uterus, other |
| Raised AST | 0.0 (0.0–18.5) | 0.0 (0.0–13.2) | 0.0 (0.0–6.1) | 4.1 (0.5–14.0) | 5.3 (0.1–26.0) | 37.5 (8.5–75.5) | **Ovary,** liver, other, lymphoma, CUP |
| Low ferritin | 0.7 (0.1–2.7) | 0.9 (0.3–2.4) | 1.8 (0.4–5.2) | 1.4 (0.0–7.5) | 7.7 (1.6–20.9) | 5.9 (1.2–16.2) | Bowel, breast, cervix |
| Raised CRP | 0.7 (0.1–2.4) | 1.8 (0.8–3.4) | 5.5 (3.7–7.9) | 9.4 (6.9–12.5) | 9.5 (6.5–13.2) | 8.7 (5.5–13.0) | **Ovary,** bowel, lung, CUP, pancreas, lymphoma, renal tract, breast, gastro-oesophageal, uterus, sarcoma |
| Raised ESR | 0.9 (0.2–2.3) | 1.1 (0.5–2.1) | 2.9 (1.9–4.2) | 6.3 (4.7–8.2) | 7.3 (5.3–9.7) | 7.0 (4.6–10.0) | **Ovary,** bowel, pancreas, lung, CUP, breast, sarcoma, renal tract, cervix, other, gastro-oesophageal, lymphoma |
| Raised total WBC | 1.8 (0.4–5.3) | 2.4 (0.9–5.1) | 5.1 (2.6–9.0) | 9.9 (5.8–15.5) | 12.0 (7.0–18.8) | 7.1 (3.3–13.1) | **Ovary,** CUP, bowel, lung, pancreas, cervix, liver, renal tract |
| Raised ALT | 0.0 (0.0–2.2) | 1.7 (0.6–3.7) | 1.8 (0.8–3.5) | 5.4 (3.3–8.4) | 7.9 (4.2–13.5) | 9.5 (3.6–19.6) | Ovary, pancreas, bowel, liver, CUP, gastro-oesophageal, lung, breast, other, sarcoma |
| Raised ferritin | 0.0 (0.0–33.6) | 0.0 (0.0–26.5) | 9.5 (1.2–30.4) | 5.6 (0.1–27.3) | 12.5 (1.6–38.3) | 8.3 (0.2–38.5) | **Ovary, sarcoma,** CUP |
| Low total WBC | 0.0 (0.0–2.7) | 0.5 (0.0–2.8) | 1.1 (0.1–3.8) | 0.7 (0.0–3.8) | 1.4 (0.0–7.3) | 3.8 (0.5–13.0) | Ovary, myeloma, leukaemia, other haem, renal tract, lung, bowel |
| Low platelets | 1.5 (0.0–7.9) | 1.2 (0.0–6.7) | 1.2 (0.0–6.4) | 2.6 (0.3–9.2) | 6.6 (2.2–14.7) | 3.4 (0.4–11.9) | Renal tract, cervix, leukaemia, other haem, liver, other, breast, lung |
| Raised bilirubin | 0.0 (0.0–3.3) | 2.0 (0.4–5.8) | 1.7 (0.2–5.9) | 4.5 (1.2–11.1) | 4.8 (1.3–11.7) | 7.3 (2.0–17.6) | CUP, breast, liver, cervix, other, lung, pancreas, bowel |
| Raised creatinine | 0.0 (0.0–70.8) | 0.0 (0.0–30.8) | 12.5 (1.6–38.3) | 4.1 (0.5–14.0) | 7.1 (2.9–14.0) | 4.3 (2.0–8.1) | **CUP, ovary,** lymphoma, bowel, uterus, other haem, melanoma, breast, liver, other, pancreas |
| Raised HbA1c | 1.3 (0.0–7.0) | 1.2 (0.2–3.4) | 1.0 (0.3–2.6) | 3.0 (1.6–5.0) | 5.3 (3.4–7.9) | 6.0 (3.4–9.7) | Ovary, pancreas, breast, bowel, CUP, lung, cervical, gastro-oesophageal, lymphoma, leukaemia, sarcoma, renal tract, uterus |
| Raised CA125 | 8.3 (1.8–22.5) | 7.0 (3.4–12.6) | 22.1 (13.9–32.3) | 50.0 (38.1–61.9) | 49.0 (34.4–63.7) | 32.4 (17.4–50.5) | **Ovary, sarcoma, CUP, bowel,** breast, pancreas, uterus, lymphoma, lung, other, cervix, liver, gastro-oesophageal, myeloma, renal tract |

Red = PPV of ≥10%; orange = PPV of ≥3% (NICE threshold for recommending cancer investigation); yellow = PPV ≥2%. Cancer sites in bold have PPVs >1%; all listed cancer sites have PPVs >0.1%; CUP, cancer of unknown primary; other haem = other haematological cancer, CI, confidence interval; CRP, c reactive protein; ESR, erythrocyte sedimentation rate; WBC, white blood cell count; PSA, prostate specific antigen.

bloating the risk of cancer exceeds the currently proposed 3% risk threshold for which urgent referrals are recommended. In patients aged 30 to 59 years with these symptoms certain blood test abnormalities strongly predict risk of any undiagnosed cancer; these include anaemia, low albumin, raised platelets, abnormal ferritin, and raised inflammatory markers. In many age-sex strata, the presence of these abnormalities raised the probability of undiagnosed cancer to above 3%. Results from readily available, commonly used blood tests in primary care enhanced the assessment of underlying cancer risk in patients presenting with these 2 nonspecific

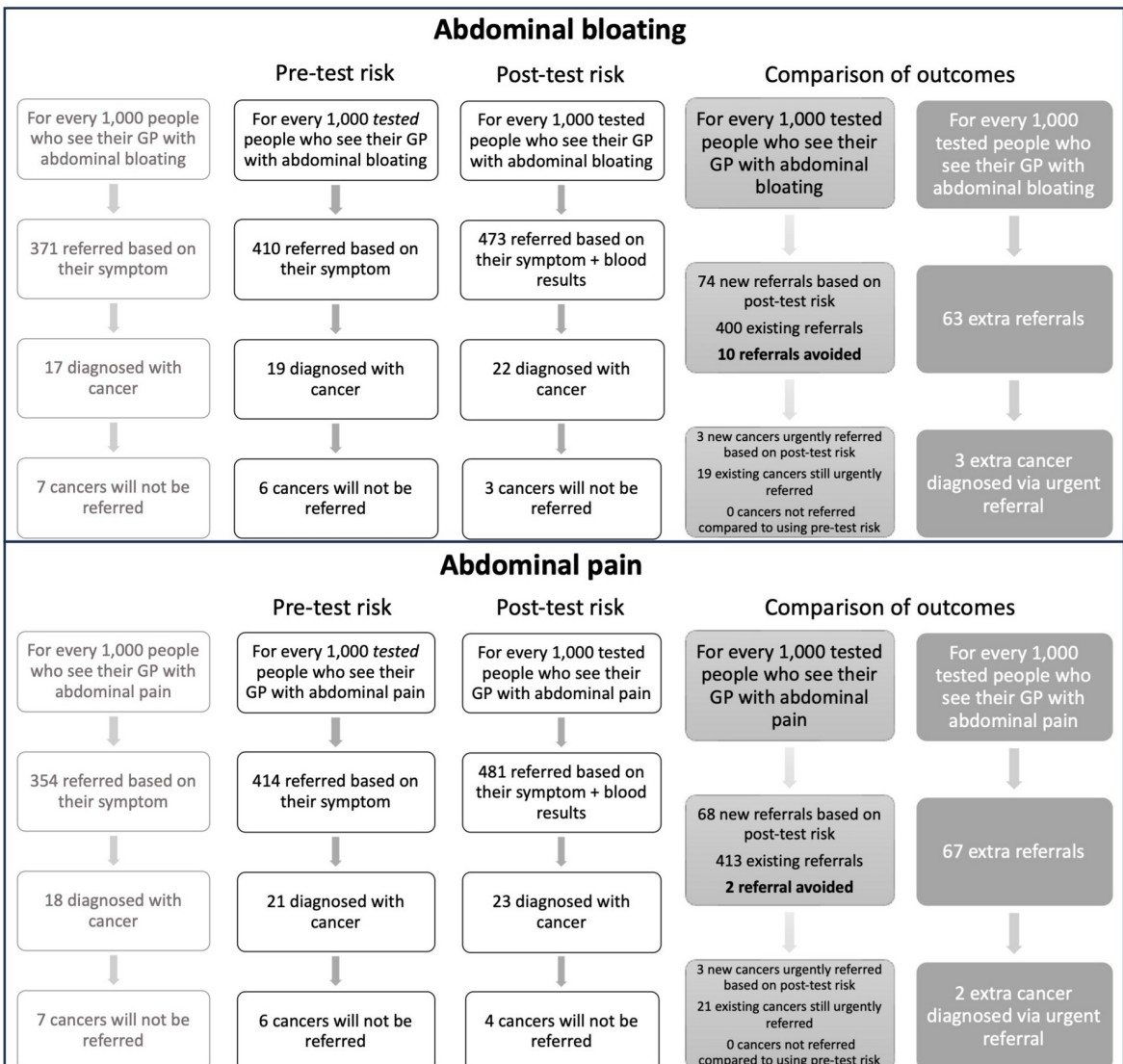

**Fig 3. Box 1: Flow chart of 1,000 patients with abdominal bloating or pain modelling who would be recommended for an urgent cancer referral based on a pretest versus posttest risk of cancer >3%.** Column 1 shows referral recommendations for our study population based on symptom (+ age and sex), applying the PPVs from Table 2; Column 2 shows recommendations based on symptom (+ age and sex) for patients in the study population who had a blood test around the time of abdominal pain/bloating presentation, applying the PPVs from Table 2; Column 3 shows updated referral recommendations for the tested study population based on symptom + blood test results (+ age and sex), applying the PPVs from Tables 3–6; Column 4 shows the number of study patients who will experience a change in recommendation based on applying posttest risk (Yes/No) versus pretest risk (Yes/No) (note as estimates are presented per 1,000 patients and rounded to whole numbers these figures will not exactly correspond to the numbers in columns 2 and 3); Column 5 shows the potential overall change in referral numbers and cancer diagnoses via urgent referral route using posttest risk (derived by comparing numbers from columns 2 and 3).

abdominal symptoms compared to using information on presenting symptom, age and sex alone. Incorporating these blood tests into referral recommendations could help to better utilise data from existing blood test results and detect additional patients with underlying cancer via expedited routes. Additionally, by ranking cancer sites according to risk for each blood test abnormality, the findings can help guide investigation strategies post-referral.

The use of linked primary care data is a key strength of this study. In healthcare systems where GPs play a gatekeeping role, primary care is where most symptomatic patients with cancer first present [47]. CPRD provides a large, representative sample of the UK population, with our study providing updated evidence from over 400,000 patients with abdominal pain and 50,000 patients with abdominal bloating linked to national cancer registry data. This enabled evaluation of the predictive value of several blood test abnormalities in patients with abdominal pain or bloating, stratified by age and sex. While the use of such electronic health record data is one of the best ways to study risks following symptomatic presentations, it does have some important weaknesses. In particular, there are often problems with the completeness and accuracy of data recording. With that said, primary care blood test results are electronically transmitted directly to the patient record from the laboratory ensuring completeness and accuracy, and cancer diagnoses in this study were captured from the English national cancer registry dataset, which is the gold standard for identification of cancer cases. Symptom recording may be less complete; comprehensive code lists for abdominal pain and bloating, developed using established methods [48,49] and cross checked by 3 clinicians, were used to identify symptomatic patients in CPRD. However, recording is reliant on coding by GPs in patients who present with a symptom, which is known to have wide variability [50,51]. As a result, the estimated PPVs are for patients who attend primary care with symptoms and where the GP considered the symptom relevant enough to be coded, rather than all instances of abdominal pain or bloating in the general population. As abdominal bloating is less common than abdominal pain, for some blood tests the number of abdominal bloating patients tested is small. These 2 symptoms have different aetiologies, management and cancer site risk profiles [6] and were therefore considered separately.

Another strength is that this study included all cancers, reflecting the broad spectrum of cancers associated with abdominal pain and bloating [6], and presented cancer site-specific PPVs for each blood test abnormality. This allowed identification of the cancer sites most strongly associated with each abnormal result. By providing rankings of the cancer sites most likely in patients with each blood test abnormality, the findings enable clinicians to prioritise investigations for malignancies in order of risk after referral. Blood tests occurring around the time of symptomatic presentation were included. This increased the likelihood that they were related to the symptom of interest and provided a real-life reflection of the information available at the time of symptom assessment [52]. The novel study design also avoids introduction of immortal time bias (where follow-up includes a period after symptom presentation where patients cannot get cancer) [53].

While our study reflects real-world use of blood tests, a key limitation is that the predictive value of blood tests abnormalities presented in this study relates to patients who have already been selected by their GP to have blood tests. The findings of this and previous studies have shown that patients selected for testing are more likely to have cancer than untested patients [22,23,54–56]. This is possibly due to them being older or having a more concerning presentation of the symptom, which cannot be detected from the coded data. The estimates presented in this study therefore apply to patients who are already selected for blood testing in primary care and will be lower in the broader population of all patients presenting with abdominal pain or bloating. We are therefore not recommending an increase in the use of blood tests in primary care, but rather better use of existing results from blood tests in the patient population selected for testing. Primary care blood testing rates may also vary over time and in different countries/healthcare settings [57]. Increased testing may also occur as a potential unintended consequence of increased inclusion of blood test results in national cancer guidance. If testing rates are higher and more patients are tested, the pretest risk of cancer in these less selective tested populations will be lower (if cancer incidence remains stable). The predictive value of

blood test abnormalities for cancer in these lower risk populations will therefore be reduced [58]. As a result, caution is needed when translating our findings to other settings or time-frames with different rates of primary care blood tests, and periodic re-estimation of the PPVs from this study will be needed to take into account possible changes in blood test use over time. Although this study has significant international relevance, the findings may not be generalisable to healthcare systems where financial or structural barriers to accessing healthcare may exist or where the role of primary care is different. The estimates of potential changes in urgent referral numbers, and subsequent cancers detected, assume that proposed referral guidelines would be fully adhered to and therefore represent the maximum potential impact of including blood test results. This will however be lower in reality, as adherence to referral guidelines for suspected cancer is suboptimal in clinical practice [59,60], which would result in fewer urgent referrals and potentially fewer additional cancers detected than estimated.

A few previous studies have examined PPVs for cancer following abdominal pain or bloating presentations in general practice. Herbert and colleagues [6] used The Health Improvement Network (THIN) UK database to examine PPVs for any cancer in the 12 months post-consultation, reporting PPVs of 1.2% to 1.3% in females and 1.7% to 1.8% in males. The 3% cancer risk threshold was exceeded in male patients aged ≥70 years, but not in females in any age-strata. Price and colleagues [7] conducted a similar study using CPRD and cancer registry data, focusing on the risk of any intra-abdominal cancer in abdominal pain patients aged ≥40. They reported PPVs up to 2.3% in females and 3.4% in males aged ≥70. Our study expands on these findings by presenting age and sex-specific PPVs in patients with either abdominal pain or bloating for underlying cancer, overall and by site. We found that PPVs for cancer exceed 3% in patients of either sex presenting with either of these symptoms aged ≥60. The higher PPVs reported in our study reflect the use of cancer registry data, which more accurately captures cancer diagnoses than primary care data [61], and inclusion of all cancer sites.

Hamilton and colleagues [62] examined the predictive value of anaemia in abdominal pain patients for colorectal cancer in the following year, reporting PPVs of 7% for abdominal pain and >10% for abdominal tenderness. Building on this, Price and colleagues [7] examined the predictive value of anaemia and raised platelets in abdominal pain patients for groups of intra-abdominal cancers, reporting PPVs for uterus or oesophagogastric cancer in females of 7% to 18% following anaemia and 3% to 6% following raised platelets. Our study fills an important gap in the literature by presenting the predictive value of a range of co-occurring blood test abnormalities in patients presenting with abdominal pain and bloating for cancer overall and by providing cancer site-specific estimates to help prioritise malignancies for investigation.

The findings of this comprehensive consideration of the value of 19 abnormal blood test results for improving detection of underlying cancer in patients with 2 nonspecific abdominal symptoms have implications for clinical practice and health policy. Existing UK NICE guidelines for urgent cancer referral in patients with abdominal pain and bloating currently only recommend referral in patients with additional clinical features [9]. We found in patients of both sexes aged 60 years or over presenting with either of these symptoms, PPVs for cancer exceed the NICE risk threshold of 3%, and a potential cancer diagnosis and further urgent assessment should be considered in these patients regardless of the presence of other symptoms or blood test results. Detection of abnormal blood test results in this age group will increase clinician confidence in the decision for further cancer assessment, thereby increasing adherence to referral guidelines as it further strengthens the pretest symptom-based assessment of cancer risk. This is important as adherence to cancer referral recommendations is currently suboptimal, even when alarm (red flag) symptoms are present [59,60].

Existing guidelines also predominantly focus on older patients, despite cancer diagnosis being more challenging in patients under 60 and this being the group more likely to experience

diagnostic delays [63]. Our findings show that existing results from readily available primary care blood tests could be better used to improve triaging and risk stratification of patients with abdominal pain and bloating in primary care, especially patients aged 30 to 59 where there is uncertainty regarding onward management because pre-blood test risk is below or near the 3% risk threshold above which urgent specialist assessment for suspected cancer is recommended. Patients with abnormal blood results, and high posttest risk of underlying cancer, could be identified and prioritised for referral, and those with normal results and low posttest risk could be monitored in primary care. As primary care clinicians are already selecting patients with abdominal pain and bloating who are higher risk of serious disease for testing [58] and around two thirds cancer patients with these symptoms are known to have routine blood tests done in primary care [16], we are not proposing an increase in blood test use. Rather, the findings present an efficient and affordable way to identify those at highest risk of underlying cancer using information from existing blood tests. These findings have the potential to substantially advance current clinical practice and inform changes to national guidelines to improve early cancer diagnosis. In addition to anaemia and platelets, which are established risk factors for underlying cancer [21,22,64,65], additional blood test abnormalities were identified which in many age-sex strata had a cancer risk exceeding 3% in patients with these symptoms and should be considered for inclusion in clinical guidelines for urgent cancer referral. These include low albumin and ferritin, raised acute phase reactants (ferritin, ESR, CRP, WBC), and abnormal bone profile tests (raised ALP or calcium). No single blood test had sufficient rule-out value for cancer based on a normal result. While the findings relate to a UK setting, the clinical presentations examined are ubiquitous across the world and the blood tests considered are readily affordable and available globally, so the findings have international relevance for clinicians, policy makers, and researchers.

We report the relative risk of cancer at different sites (presented as the rank order of site-specific PPVs). Current international cancer guidelines relating to abdominal pain and bloating predominantly focus on risk of specific cancer sites, principally colorectal cancer [9,14]. However, these nonspecific abdominal symptoms are presenting features of several cancer sites [6]. Although ovarian, colon, and pancreatic cancers were most common, the range of possible cancer sites was diverse. Primary care investigation of patients presenting with these symptoms should therefore consider the possibility of more than 1 cancer site, for example, not just using CA125 to assess the risk of ovarian cancer in females with abdominal bloating but also potentially considering a FIT test (for suspected colorectal cancer). By identifying cancer sites at highest risk, abnormal blood test results could help support selection of appropriate clinical specialties for referral and guide optimal testing strategies pre- and post-referral (including in Rapid Diagnostic Centres) [9,66,67].

The study findings have potential health system implications. Inclusion of blood test results in cancer referral clinical guidelines could have the unintended consequence of increasing primary care blood test requests, which would place an additional burden on phlebotomy and primary care services. Additionally, as modelled in this study, the expected increase in cancer patients detected through urgent referral pathways will come at the cost of increased secondary care referrals and specialist investigations. Consistent with most available evidence in our field and NICE guideline recommendations that do not incorporate consideration of stage at diagnosis, we have not stratified by stage at diagnosis. However, as cancers diagnosed through urgent referral routes have substantially higher 1-year survival [47] and diagnostic intervals are shorter for patients on urgent pathways [68,69], we can posit that if the implementation of our study findings result in a greater number of cancers diagnosed through such referrals, this will also result in more cancers being diagnosed at an earlier stage improving prognosis. Formal modelling of cancer stage is however needed to confirm this alongside health economic

analysis to evaluate health system effects and ensure appropriate infrastructure and resources are in place.

It should be noted that blood test abnormalities are common in primary care and we showed that the predictive value of blood tests in symptomatic patients for detecting cancer varies by age, sex, and test type. It may therefore be challenging for clinicians to identify relevant abnormalities in practice unless supported by automated clinical decision support tools integrated into existing IT systems to help process and interpret the wealth of information in blood test data [70]. The predictive accuracy of individual blood test abnormalities examined in this study may also be further enhanced by combining effects from different blood test results, their trends over time, and other clinical features using multivariable prediction models [23,30,52]. The extent that multiple abnormal tests will improve the PPV beyond the value of a single abnormal test is not known and needs to first be established empirically, accompanied by evidence on how to implement these models and ensure their uptake in clinical practice [71]. The current NICE guidelines operate on the basis of simpler clinical scenarios characterised by the presence/absence of presenting features. Our study findings of various symptom-single test combinations that can reclassify some patients aged 30 to 59 who present with abdominal pain and bloating as having a risk of underlying cancer >3% could inform updates to these clinical guidelines, with potential for further improvement in the future as evidence emerges. Finally, we focused on 2 nonspecific abdominal symptoms in this study which pose a diagnostic challenge in primary care and are associated with risk of different cancer sites [6]. It is likely consideration of blood test data will also enhance assessment of cancer risk for other nonspecific symptoms [52,72], which should be explored in follow-on research.

## Conclusions

Results from existing commonly used blood tests can substantially improve the clinical triage of patients presenting with abdominal pain and bloating by identifying those at highest risk of underlying cancer who should be considered for urgent referral. The findings support the development of guideline recommendations, particularly in patients aged 30 to 59 with the studied symptoms who have a baseline risk of undiagnosed cancer below referral thresholds, where consideration of concurrent blood test abnormalities can enable specialist assessment where warranted, and identify the likely cancer sites relating to blood test abnormalities to guide onward investigation and referral strategies. This would enable the expedited investigation, referral, and diagnosis of patients with these symptoms who are most likely to have cancer.

## Supporting information

**S1 Supplementary File.** Table A. Read codes for abdominal pain and bloating. Table B. Evidence and rationale for blood tests included in the study. *readterms assigned to each code were manually reviewed by a clinician and irrelevant terms were removed. Table C. ICD-10 codes for Cancer. Fig A. Creation of the final tested symptomatic population by combining 2 subgroups of patients who had a blood test request around the time of abdominal symptom presentation identified by 2 different presentation scenarios. Fig B. Monthly cancer incidence following new GP presentation of abdominal pain or abdominal bloating. Statistically estimated inflection point is 10 months post symptom presentation. Table D. Risk ratios (observed/expected) for cancer in the 12 months following abdominal pain or bloating compared to expected rates in the population. Population rates use ONS cancer incidence and population size estimates for mid-2011; (excess risk = observed risk-expected risk). Table E. Predictive value for cancer diagnosis by sex at different time points after presenting to primary care with abdominal pain or bloating. Males, $n = 156,590$ for abdominal pain and $n = 14,255$

for abdominal bloating; females, *n* = 268,959 for abdominal pain and *n* = 38,066 for abdominal bloating. PPV, positive predictive value; m, month. Table F. Breakdown of cancer diagnosis by site and sex in patients presenting to primary care with abdominal pain or bloating who develop cancer in the following 12 months. Top section = males (*n* = 4,372 for abdominal pain and *n* = 367 for abdominal bloating). Bottom section = females (*n* = 5,055 for abdominal pain and *n* = 781 for abdominal bloating); *P*-value from chi squared test. Table G. Blood test use in the 3 months pre/post GP presentation with new onset abdominal pain or bloating in patients with ≥1 blood test, by sex. SD, standard deviation; IQR, interquartile range; *for 16 prespecified blood tests in the 3 months before the index date. Table H. Baseline characteristics of patients presenting to their GP with new onset abdominal pain with a blood test in the 3 months pre versus post presentation. Total *n* = 273,155; patients can belong to both groups; *P*-value compares both cohorts using chi squared test, SD, standard deviation; IQR, interquartile range; *for 16 prespecified blood tests. Table I. Baseline characteristics of patients presenting to their GP with new onset abdominal bloating with a blood test in the 3 months pre versus post presentation. Total *n* = 36,633; patients can belong to both groups; *P*-value compares both cohorts using chi squared test, SD, standard deviation; IQR, interquartile range; *for 16 pre-specified blood tests. Table J. Positive predictive values (PPVs) for cancer in 12 months by sex in patients with abdominal pain and a blood test abnormality. Red = PPV of ≥10%; Orange = PPV of ≥3% (NICE threshold for recommending cancer investigation); Yellow = PPV ≥2%. *n*/*N* = number of patients with an abnormal test/all patients tested; LR, likelihood ratio; TM, tumour markers (PSA or CA125). Table K. Positive predictive values (PPVs) for cancer in 12 months by sex in patients with abdominal bloating and a blood test abnormality. Red = PPV of ≥10%; Orange = PPV of ≥3% (NICE threshold for recommending cancer investigation); Yellow = PPV ≥2%. *n*/*N* = number of patients with an abnormal test/all patients tested; LR, likelihood ratio; TM, tumour markers (PSA or CA125). Table L. Positive predictive values (PPVs) for different cancer sites within 12 months by sex in tested patients with abdominal pain or bloating and a blood test abnormality: males with abdominal pain, females with abdominal pain, males with abdominal bloating, females with abdominal bloating. PPVs >10% highlighted in red, >3% highlighted in orange, >2% highlighted in yellow, >1% highlighted in blue. Table M. Risk ratios (RR) comparing 1 year cancer risk in patient with abdominal pain and a normal versus abnormal test result for 19 different blood test abnormalities. *adjusted for age. Table N. Risk ratios (RR) comparing 1 year cancer risk in patient with abdominal bloating and a normal versus abnormal test result for 19 different blood test abnormalities. *adjusted for age. Study analysis plan.
(DOCX)

**S1 STROBE Checklist. STROBE Statement—Checklist of items that should be included in reports of observational studies.**
(DOCX)

# Acknowledgments

We thank all patients and general practices who shared their data to make this research possible. The interpretation and conclusions contained in this study are those of the author/s alone.

# Author Contributions

**Conceptualization:** Meena Rafiq, Cristina Renzi, Georgios Lyratzopoulos, Matthew Barclay.

**Data curation:** Meena Rafiq, Becky White.

**Formal analysis:** Meena Rafiq.

**Investigation:** Meena Rafiq, Becky White, Brian Nicholson, Georgios Lyratzopoulos.

**Methodology:** Meena Rafiq, Cristina Renzi, Georgios Lyratzopoulos, Matthew Barclay.

**Supervision:** Cristina Renzi, Georgios Lyratzopoulos, Matthew Barclay.

**Visualization:** Nadine Zakkak.

**Writing – original draft:** Meena Rafiq, Georgios Lyratzopoulos, Matthew Barclay.

**Writing – review & editing:** Meena Rafiq, Cristina Renzi, Becky White, Nadine Zakkak, Brian Nicholson, Georgios Lyratzopoulos, Matthew Barclay.

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
