## [Editor Report · Decision Letter 0]

16 Jan 2024

Dear Dr Rafiq, 

Thank you for submitting your manuscript entitled "Predictive value of abnormal blood tests for detecting cancer in primary care patients with non-specific abdominal symptoms: A population-based cohort study of 477,870 patients in England" for consideration by PLOS Medicine.

Your manuscript has now been evaluated by the PLOS Medicine editorial staff and I am writing to let you know that we would like to send your submission out for external peer review.

Please re-submit your manuscript within two working days, i.e. by Jan 18 2024 11:59PM.

Feel free to email me at pdodd@plos.org or the team at plosmedicine@plos.org if you have any queries relating to your submission.

Kind regards,

Pippa

Philippa Dodd, MBBS MRCP PhD

PLOS Medicine

---

## [Decision Letter · Decision Letter 1]

4 Mar 2024

Dear Dr. Rafiq,

Many thanks for submitting your manuscript "Predictive value of abnormal blood tests for detecting cancer in primary care patients with non-specific abdominal symptoms: A population-based cohort study of 477,870 patients in England” (PMEDICINE-D-24-00134R1) to PLOS Medicine. The paper has been reviewed by three subject experts and a statistician; their comments are included below and can also be accessed here: [LINK]

As you will see, the reviewers were positive about the paper but, they raised a number of questions about specific study details and the methodological approach. After discussing the paper with the editorial team and an academic editor with relevant expertise, I’m pleased to invite you to revise the paper in response to the reviewers’ comments. We plan to send the revised paper to some of all of the original reviewers*, and of course we cannot provide any guarantees at this stage regarding publication.

When you upload your revision, please include a point-by-point response that addresses all of the reviewer and editorial points, indicating the changes made in the manuscript and either an excerpt of the revised text or the location (eg: page and line number) where each change can be found. Please submit a clean version of the paper as the main article file and a version with changes marked should as a marked-up manuscript. Please also check the guidelines for revised papers at http://journals.plos.org/plosmedicine/s/revising-your-manuscript for any that apply to your paper.

We ask that you submit your revision by March 25th 2024. However, if this deadline is not feasible, please contact me by email, and we can discuss a suitable alternative.

Please don’t hesitate to contact me directly with any questions (pdodd@plos.org). If you reply directly to this message, please be sure to ‘Reply All’ so your message comes directly to my inbox.

Kind regards,

Pippa

Philippa Dodd, MBBS MRCP PhD

PLOS Medicine

plosmedicine.org

pdodd@plos.org

*Please note: If your article is accepted, you may have the opportunity to make the peer review history publicly available. The record will include editor decision letters (with reviews) and your responses to reviewer comments. If eligible, we will contact you to opt in or out.

Editorial comments:

1) We enjoyed reading your paper and agree that it offers some valuable insights. However, we also agree with the reviewers that the lack of methodological detail included make it difficult to understand and that further consideration should be given to the relevance in clinical practice and implications for healthcare systems. Please revise your manuscript in light of reviewer and Academic Editor comments detailed below.

2) Did your study have a prospective protocol or analysis plan? Please state this (either way) early in the Methods section.

Comments from the reviewers:

Reviewer #1: TITLE & ABSTRACT

1. The abstract should specify all types of data used - please make reference to the HES APC and IMD databases here.

2. Were there any prespecified hypotheses? If so, these should be stated in the objectives.

METHODS

1. Were any patients censored before 12 months (potentially due to death or migration)? If so, how was their follow-up managed?

2. Was any validation conducted of the codelist for abdominal pain and bloating?

3. Please include the codelist for the included blood tests.

4. Please describe the extent to which you had access to the database populations used to create the study population. If this was conducted by others, please include them in the acknowledgements.

5. Please confirm that the study included person-level data linkage.

RESULTS

1. The selection of included patients, including filtering based on data quality and linkage, should be described in detail. This would be best placed in the study flow diagram.

2. Assuming censoring occurred, please summarise the average and total amount of follow-up time

DISCUSSION

1. Please provide some discussion on the potential cost-effectiveness of changes to guidance.

OTHER INFORMATION

1. Please provide information on how to access the study protocol and programming code.

Reviewer #2: This research on abdominal pain/bloating, blood tests and PPV for cancers by age and sex has significant implications for general practice. 

Comments:

In abstract, methods and findings (line 12): The authors state PPV, NPV, sensitivity and specificity were calculated; however, the focus was on PPVs.

Introduction

(page 4, line 11): It would be good to understand what the NICE guidelines recommend. How urgent do the referrals need to be made? How urgent do the patients need to be seen by the referred?

Methods 

(page 5, line 12-15): Some context on the CPRD and English primary care health system would be useful to the international readership.

Results 

(page 8, line 28-29): "Cancer incidence was higher in males….." The authors are referring to PPV for cancer, not cancer incidence. It may also be worthwhile rearranging the sentence to replace "or" with another conjunction.

(page 12, line 30 and elsewhere): The authors use "cancer risk >2%" to refer to "positive predictive value". These are different concepts. 

(page 13, line 12):"…the predictive value of blood test abnormalities was lower…". Could the authors be possibly referring to "…the PPV for cancers associated with blood test abnormalities…"? 

Discussion 

(page 19, line 3-4 and page 19, line 25-26): Could the authors be possibly referring to the PPV for cancers again?

(page 20, line 6-7): It would be helpful to know how GP decisions around blood testing could impact on PPVs. 

(page 22, line 18-19): The PPVs for cancers would be helpful for triaging; however PPVs and cancer risks are different concepts. There may be a better term than "risk stratification". 

Reviewer #3: This is an interesting paper with clear background, mostly comprehensive description of methods and results, with plenty of relevant supplementary data. I enjoyed reading the manuscript. Due to the amount of information, at times I struggled a bit to understand the key message from the paper, and what research is still needed if blood tests are to be used in primary care to triage patients. Would the team recommend all the investigated tests, for all aged 30-59, with a suspicion of (any) cancer? Further information on the clinical and practical implications of this would be useful. Therefore, I would recommend further revisions to the manuscript.

Major Issues

1. Further clarifications on the model would be useful:

1.1. Methods, page 8, line 14 - I presume that 100% guideline concordance was used as this is the gold standard, but is the gold standard realist? (see https://bjgp.org/content/66/643/e106)/ Would results change if estimated concordance was closer to real-life practice? This is briefly approached in the discussion, but the model was only done to show the "maximum potential impact"

1.2. Results, page 18, lines 1-9: More information on the clinical relevance of the results would be useful. Using the tests would result in many more referrals, and a few extra cancers. At an individual level, one would want their cancer diagnosed despite resource implications. What about from a clinical perspective (both primary and secondary care)? Because the model is also for any cancer and is a hypothetical model, there is no information on stage at diagnosis/tumour aggressiveness, what age groups would benefit, or whether these cancers would have been identified anyway, or later on (without impact on outcomes). Therefore, discussing possible clinical implications would be very helpful.

1.3. Results, Box 1. Further clarifications on how these calculations were made would be quite helpful. For example, for abdominal bloating we have 371 (symptom+age+sex model), 410 (as before + selected for test), and 473 (post-test risk, i.e as before + test results). How do these numbers result in 74 new referrals, 400 existing referrals and 10 referrals avoided? Furthermore, I am not sure if the number of extra referrals in the final column is the difference between the pre-test and the post-test risk, or the difference between symptom+age+sex (i.e. first column) and post-test risk? Methods seem to indicate the latter, but I am unsure of it based on the results. Finally, why is pre-test risk shown on top of the second column for abdominal bloating, but is the title for both column 1 and column 2 for abdominal pain? 

2. Discussion, page 20, line 18 - The authors raise the challenge of spectrum bias/effect here. It may be worth adding a caveat to the discussion: before using these blood tests for triage, or deciding the best way to use these tests in primary care, it is important that prospective studies are carried out in primary care so estimates are accurate for this population. What do authors think should be done next (i.e. recommendations for future research)?

3. Several tables in supplementary data: When each cancer site is looked at separately, we can then see that the value of tests (and different tests) vary a lot by cancer site. The PPVs are quite low for many cancers. The very high PPVs are for specific tumour types and specific tests e.g. ovarian cancer for raised CA125 or raised PSA and prostate cancer. What does that tell us in terms of the benefits of using all the studied blood tests, for all tumour types? Could the authors explore further the clinical implications, and the applicability of the results?

Minor issues

4. Results, Table 1: There is a note about the blood tests being ordered for 15-pre-specified symptoms - could this be clarified? Is this this about the medical indication (in general) for such tests?

5. Results, page 9, lines 1-3. For those aged 60+ PPVs for either symptom already resulted in PPV above the NICE referral threshold. What would be the main benefit of such tests for this age group? Would easily accessible blood tests give confidence to the professional that this patient should be tested further? 

6. Could the authors justify why they chose to focus on abdominal pain and bloating among other possible non-specific symptoms? I understand abdominal pain is a non-specific symptom that can occur in many symptomatic presentations that may indicate different types of cancer (particularly GI cancers), but bloating is less common (although a more recognised risk for ovarian cancer). Since almost all cancer types were included, why these two?

7. The study included patients aged ≥30. Cancer risk increases with older age, so it makes sense not to include younger populations. However, could the chosen threshold be justified?

8. Results, Table 1 - It would have been interesting to have one line for male sex and another for female sex, particularly to see differences in testing for bloating

9. Table 3a is interesting and has a lot of information. It is good to see the cancer sites listed from highest to lowest risk. Is there any way to highlight amongst these the ones with a PPV of 3% and higher (maybe using the same colours as in the age cells (i.e. orange or red) for the font)?

10. Conclusion section: Authors mention that the results from blood tests can substantially improve the clinical triage of patients, but no NPVs (very important is this scenario) were discussed (although they are available in Supplementary Table S9a and S9b). As before, spectrum bias is an issue. Do we have sufficient evidence to write the suggested guidelines for those aged 30-59?

11. Discussion, page 22, lines 1-4. This paragraph talks about using the results to help prioritise malignancies. Could this be clarified further? How would this be done?

12. Supplementary Table S10 and S11: I understand that the focus was on PPV due to the threshold for urgent referral, but NPVs may be more relevant here if the aim is to triage in primary care (as mentioned in the conclusion). It would also be helpful to highlight the PPVs over 3%, as they show which blood tests are more relevant here, for which tumour type (e.g. anaemia, low albumin and low ferritin for bowel (male), PSA for prostate, and CA 125 for ovarian cancers.

Miscellaneous

13. Results, Table 1 - Numbers for 2015-2016 seem to be much smaller compared to other years (even when considering that data are only available up to October 2016). Was there also a time lag in the data (due to delays in CRPD receiving primary care information) when extraction happened?

14. Figure 2 is a bit blurred, but I checked the original submission and it is much more visible so it may have just been an issue with resubmission. I would suggest replacing "abdominal symptoms" in the title with "abdominal pain/bloating". 

15. Supplementary Figure S2: Consider making it clear that the "cohorts" here refer to the "Scenarios" described in the paper

Reviewer #4: This is an interesting study on the predictive value of abnormal blood tests for detecting cancer in primary care patients with non-specific abdominal symptoms using the UK CPRD data linked to the National Cancer Registry. However, there are a few major issues needing attention.

1. The result and interpretation of PPV and NPV are difficult to follow. In a typical predition study, we would expect PPV and NPV to be 80%-90% so that the rules can be applied in practice. However, here in the paper, we are talking about 2-3% then improved to 8%-10% when add the information of the blood tests. It's obvious that the PPV will improve when add more informtion such as these blood tests. We know this. More importantly, it would be good to use these blood tests to bulid a comprehensive multivariate prediction model to predict either referral or incidence of cancer. So far the paper is mostly exploratory on extra information from blood test and hasn't given a solid and overall solution. 

2. The statistical analyses are a bit simplistic and mainly univariate. The combined impact of 19 blood tests would be preferred using the multivariate analysis.

3. Overall, the paper pointed out thes blood tests were useful, which is mostly known. However, it doesn't go further to build a useful and practical model to be used in the clinical settings.

[LINK]

Comments from the Academic Editor:

This is an intriguing, methodologically well-done analysis that could potentially influence clinical practice. It is also an excellent example of how integrated medical practices with large datasets can use such data to better serve their clientele. I have just a few thoughts to offer:

First, I greatly appreciate the care that the authors put into correctly determining how to count person-time and avoid immortality bias. The authors note two events that must occur before follow-up can start: 1) a blood test and 2) a symptom report of abdominal pain. The date by which both events have occurred is the index date for follow up. Logically speaking, cancer diagnoses between events should not be counted, else results could be affected by “immortality bias”.

While they appear to correctly handle this issue, there are inconsistencies in their descriptions of t

---

## [Decision Letter · Decision Letter 2]

29 May 2024

Dear Dr. Rafiq,

Thank you very much for re-submitting your manuscript "Predictive value of abnormal blood tests for detecting cancer in primary care patients with non-specific abdominal symptoms: A population-based cohort study of 477,870 patients in England" (PMEDICINE-D-24-00134R2) for review by PLOS Medicine.

I have discussed the paper with my colleagues and the academic editor and it was also seen again by 2 reviewers. I am pleased to say that provided the remaining editorial and production issues are dealt with we are planning to accept the paper for publication in the journal.

[LINK]

We look forward to receiving the revised manuscript by Jun 05 2024 11:59PM.   

Kind regards,

Pippa

Philippa Dodd, MBBS MRCP PhD 

PLOS Medicine

plosmedicine.org

pdodd@plos.org

Requests from Editors:

GENERAL

Thank you for your detailed and considered responses to previous comments. We agree with reviewer #4 (please see below) that including additional rationale within the manuscript (as in your rebuttal) regarding the chosen analytical approach would be helpful to the reader. 

Please address all further comments prior to publication.

Most editorial comments pertain to specific content and formatting requirements. Some may have already been incorporated into the manuscript and others may not apply but please review the complete list of items and ensure that each is included/addressed as necessary.

STUDY REPORTING

Thank you for including the STROBE checklist. Please amend to refer to section and paragraph numbers, rather than page numbers as the latter often change at publication.

DATA AVAILABILITY

For all submissions to PLOS in which author-generated code underpins findings in the manuscript, we require that authors make all author-generated code available without restrictions upon publication of the work. In cases where code is central to the manuscript, we may require the code to be made available as a condition of publication. Authors are responsible for ensuring that the code is reusable and well documented. 

PLOS Medicine encourages all authors to share their code in line with our commitment to Open Access Science and transparent data reporting. Please see here for further details on our code sharing policy and recommendations for sharing. 

https://journals.plos.org/plosmedicine/s/materials-software-and-code-sharing#loc-sharing-code

Authors are responsible for providing the source code needed to replicate the study's findings in a repository (such as GitHub, SourceForge or Bitbucket) or a cloud computing service (such as Code Ocean). Protection of authors’ intellectual property will not be cause for exception. Please explain in the manuscript’s Data Availability Statement how readers can access the shared code. Please note that an author cannot serve as the point of contact for requests.

STATISTICAL REPORTING

Throughout all subsections, including tables and figures, please quantify the main results with 95% CIs and p values.

When reporting p values please report as <0.001 and where higher as p=0.002, for example. If not reporting p values, for the purpose of transparent data reporting, please clearly state the reasons why not. 

When reporting 95% CIs please separate upper and lower bounds with commas instead of hyphens as the latter can be confused with reporting of negative values.

Please include the actual amounts and/or absolute risk(s) of relevant outcomes (including NNT or NNH where appropriate), not just relative risks or correlation coefficients. (example for absolute risks: PMID: 28399126).

ABSTRACT

Abstract Methods and Findings:

Please ensure that all numbers presented in the abstract are present and identical to numbers presented in the main manuscript text.

Please include the study design, population and setting, number of participants, years during which the study took place, length of follow up, and main outcome measures.

Please quantify the main results (with 95% CIs and p values).

Please include the important dependent variables that are adjusted for in the analyses.

Please include the actual amounts and/or absolute risk(s) of relevant outcomes (including NNT or NNH where appropriate), not just relative risks or correlation coefficients. (example for absolute risks: PMID: 28399126). 

Please include a summary of adverse events if these were assessed in the study.

In the last sentence of the Abstract Methods and Findings section, please describe the main limitation(s) of the study's methodology.

AUTHOR SUMMARY

At this stage, we ask that you include a short, non-technical Author Summary of your research to make findings accessible to a wide audience that includes both scientists and non-scientists. The authors summary should consist of 2-3 succinct bullet points under each of the following headings:

• Why Was This Study Done? Authors should reflect on what was known about the topic before the research was published and why the research was needed.

• What Did the Researchers Do and Find? Authors should briefly describe the study design that was used and the study’s major findings. Do include the headline numbers from the study, such as the sample size and key findings. 

• What Do These Findings Mean? Authors should reflect on the new knowledge generated by the research and the implications for practice, research, policy, or public health. Authors should also consider how the interpretation of the study’s findings may be affected by the study limitations. In the final bullet point of ‘What Do These Findings Mean?’, please describe the main limitations of the study in non-technical language.

The Author Summary should immediately follow the Abstract in your revised manuscript. This text is subject to editorial change and should be distinct from the scientific abstract. Please see our author guidelines for more information: https://journals.plos.org/plosmedicine/s/revising-your-manuscript#loc-author-summary

INTRODUCTION

Please ensure that you address past research and explain the need for and potential importance of your study. Indicate whether your study is novel and how you determined that. If there has been a systematic review of the evidence related to your study (or you have conducted one), please refer to and reference that review and indicate whether it supports the need for your study.

METHODS and RESULTS

In the manuscript text, please ensure that you have indicated: 

(1) the specific hypotheses you intended to test, 

(2) the analytical methods by which you planned to test them, 

(3) the analyses you actually performed, and 

(4) when reported analyses differ from those that were planned, transparent explanations for differences that affect the reliability of the study's results. If a reported analysis was performed based on an interesting but unanticipated pattern in the data, please be clear that the analysis was data-driven.

Please remove the word ‘retrospective’ and refer to the study as simply a population-based cohort study (page 5, line 15, for example). Please check and amend throughout all sub-sections of the manuscript and supporting files.

Please ensure that you report the number of [patients, samples, etc] and dates of recruitment, and account for all methods used in your study.

Please provide the actual numbers of events for the outcomes, not just summary statistics or ORs.

Please ensure to present numerators and denominators used to derive percentages.

Please ensure to follow the statistical reporting guidance as detailed above, reporting 95% CIs alongside p values.

When reporting p values please also provide the statistical test used to determine them.

Please also see reviewer comments below.

TABLES and FIGURES

Please ensure all tables/figures are affiliated to an appropriate title/caption/footnote which clearly describes their content without the need to refer to the text.

The current table titles are long and at times it is difficult to fully understand what the tables are showing.

Please ensure all abbreviations are clearly defined within the footnote, including those used for statistical reporting.

Please separate upper and lower bounds of 95% CIs with commas as opposed to hyphens (to prevent confusion with reporting of negative values).

Where 95% CIs are reported, please also report p values as detailed above (see under statistical reporting).

Please ensure that you indicate whether analyses are adjusted or unadjusted and, where adjusted analyses are presented, please also present the unadjusted analyses for comparison. Where relevant please indicate which factors are adjusted for.

Please ensure that you indicate for the reader the meaning the meaning of all dots, lines and bars used in the figures. Figure 2, for example, would be benefit from some additional information.

To make your figures more accessible to those with colour blindness please consider avoiding the use of green and/or red.

Please also see reviewer comments below.

DISCUSSION

Please remove all sub-headings from the discussion such that it reads as continuous prose.

Page 25, line 1 – suggest instead, ‘We report the relative risk of cancer at different sites…’

REFERENCES

For in-text reference callouts please place citations in square parentheses separate by commas. For example, [1,3,6] or [1-3]. Please check and amend throughout all sub-sections of the manuscript and supporting files.

In the bibliography, please ensure that you list up to but no more than 6 author names followed by et al.

For all web references please ensure you include an, ‘Accessed [date].’

Journal name abbreviations should be those listed in the National Center for Biotechnology Information (NCBI) databases.

SUPPORTING INFORMATION

Please cite and label your Supporting Information as outlined here: https://journals.plos.org/plosmedicine/s/supporting-information

In the published article, supporting information files are accessed only through a hyperlink attached to the captions. For this reason, you must list captions at the end of your manuscript file. You may include a caption within the supporting information file itself, as long as that caption is also provided in the manuscript file. Do not submit a separate caption file.

As above, please include the amended STROBE checklist.

Please apply all guidance detailed above to the files contained within the supporting information.

SOCIAL MEDIA

To help us extend the reach of your research, please detail any X (formerly Twitter) handles you wish to be included when we tweet this paper (including your own, your coauthors’, your institution, funder, or lab) in the manuscript submission form when you re-submit the manuscript.

Comments from Reviewers:

Reviewer #3: I believe that the authors have comprehensively addressed all my comments. I have read through all responses, and checked the manuscript again. It reads well, and its message is much clearer now. I have no further queries about it.

Please see below some very minor issues I noticed when re-reading the paper just in case the authors wish to address these. They are mostly cosmetic (if these are not appropriate I trust the Editor to remove them). I have always appreciated when similar minor issues were noted to me so I will mention them here:

1. Methods, Page 5, Line 30: You may wish to add a single quotation mark to the word 'tested' here as this is how it seems to be shown elsewhere (it may be worth doing a quick search to confirm the term is shown consistently throughout the manuscript).

2. Table 1: Notes added here end with double quotation marks (perhaps when the text was copied from the response to reviewers?). Should these quotation marks be removed?

3. Table 3a: Should the abbreviation be "haem" instead of "heam"? If so, the same issue needs to be corrected in the last column of table 3b (row for low albumin)

4. Supplementary Table S10a - A zero may be missing after 67.7 (Sensitivity for Raised ESR) as all other values have two decimal point

---

## [Editor Report · Decision Letter 3]

13 Jun 2024

Dear Dr Rafiq, 

On behalf of my colleagues and the Academic Editor, Professor Steven Moore, I am pleased to inform you that we have agreed to publish your manuscript "Predictive value of abnormal blood tests for detecting cancer in primary care patients with non-specific abdominal symptoms: A population-based cohort study of 477,870 patients in England" (PMEDICINE-D-24-00134R3) in PLOS Medicine.

Prior to publication when completing your formatting changes please address the following:

1) Author Summary – line 24, suggest,’…patients presenting to primary care with non-specific…’.

2) Statements of declaration – page 29. Please remove all statements, except for the acknowledgements and author contributions, and include the others only in the manuscript submission when you re-submit your manuscript. These will be compiled as metadata at the time of publication.

3) Supporting information – in order to ensure accurate hyperlinking of supporting information files to the main manuscript at the time of publication, we require the following:

i) File names should correspond exactly with titles and with in-text citations. For example, a PDF file for ‘S1 Appendix’ must be titled and cited as ‘S1 Appendix’ and the uploaded file named ‘S1_Appendix.pdf’.

Please amend the naming of ‘Supplementary File’ to read as, ‘S1 Supplementary File’ and name the file as ‘S1_Supplementary File.docx’.

Please ensure to rename the current title within the document i.e., as ‘S1 Supplementary File’.

Please ensure to amend in-text citations throughout the manuscript, as ‘S1 Supplementary File’.

ii) As you present all tables and figures within the S1 Supplementary File these should be re-named using alphabetical naming. For example, ‘Table A’, ‘Table B’ and so on in sequence. Similarly, you should apply the same to the figures. For example, ‘Fig A’, ‘Fig B’ and so on. 

Labels that include both a number and a letter (e.g., Table S10a and Table S10b as currently labelled) should not be renamed as Table J1 or Table J2 (or Table Ji, Table Jii) – they should be named with an individual letter for example, ‘Table J’ and ‘Table K’.

Please ensure to amend the in-text citations throughout, such that these correspond to the table/figure labels. Please cite tables and figures as ‘Table A (or ‘Fig A’) in S1 Supplementary File’, and so on.

iii) Please re-name the STROBE Checklist as ‘S2 STROBE Checklist’. As Above, please rename the uploaded file as ‘S2_STROBE Checklist.docx’.

Please ensure to amend in-text citations accordingly.

*Please also be reminded to amend all titles/captions on pages 35 and 36.

4) Throughout the main manuscript and supporting files, please ensure to define all abbreviations including those used for statistical reporting within the table/figure captions.

PRESS

Thank you again for submitting to PLOS Medicine, it has been a pleasure handling your manuscript. We look forward to publishing your paper. 

Kind regards,

Pippa 

Philippa Dodd, MBBS MRCP PhD 

Senior Editor 

PLOS Medicine

pdodd@plos.org